# Leveraging Data to Say No: Memory Augmented Plug-and-Play Selective Prediction

**Aditya Sarkar**[1,2]   **Yi Li**[3]   **Jiacheng Cheng**[2,4,†]   **Shlok Mishra**[1,5]   **Nuno Vasconcelos**[2]
[1]University of Maryland, College Park   [2]University of California, San Diego
[3]Qualcomm AI   [4]Yale University   [5]Meta AI

## Abstract

Selective prediction aims to endow predictors with a reject option, to avoid low confidence predictions. However, existing literature has primarily focused on closed-set tasks, such as visual question answering with predefined options or fixed-category classification. This paper considers selective prediction for visual language foundation models, addressing a taxonomy of tasks ranging from closed to open set and from finite to unbounded vocabularies, as in image captioning. We seek training-free approaches of low-complexity, applicable to any foundation model and consider methods based on external vision-language model embeddings, like CLIP. This is denoted as *Plug-and-Play Selective Prediction* (PaPSP). We identify two key challenges: (1) *instability of the visual-language representations*, leading to high variance in image-text embeddings, and (2) *poor calibration of similarity scores*. To address these issues, we propose a *memory augmented* PaPSP (MA-PaPSP) model, which augments PaPSP with a retrieval dataset of image-text pairs. This is leveraged to reduce embedding variance by averaging retrieved nearest-neighbor pairs and is complemented by the use of contrastive normalization to improve score calibration. Through extensive experiments on multiple datasets, we show that MA-PaPSP outperforms PaPSP and other selective prediction baselines for selective captioning, image-text matching, and fine-grained classification. Code is publicly available at https://github.com/kingston-aditya/MA-PaPSP.

## 1 Introduction

The success of vision-language models (VLMs) enables a wide range of promising applications. However, these models are prone to erroneous predictions for reasons that include incorrect alignment of the two modalities, image or language ambiguities, or samples from the tails of their training distribution. This hampers their usefulness for real-world applications that require performance guarantees. As shown in Figure 1, *selective prediction* (SP) (Chow, 1957; 1970; El-Yaniv & Wiener, 2010; Whitehead et al., 2022; Dancette et al., 2023; Geifman & El-Yaniv, 2017; Wang & Vasconcelos, 2018; Srinivasan et al., 2024) methods address this problem by *refusing* to make low confidence predictions, with the goal of minimizing the prediction *risk* on the samples that they *accept*. Optimal selective predictors optimize the trade-off between *risk* and *coverage*, which is the acceptance ratio throughout a dataset. While there is a literature in SP, the problem has mostly been considered for *closed-set* prediction, usually with finite and small output vocabularies. Examples include classification with finite labels (Wang & Vasconcelos, 2018; Wu et al., 2020) or visual question answering from a finite set of choices (Wang et al., 2023; Dancette et al., 2023), *e.g.* binary questions, questions involving a finite set of attributes, etc. However, vision-language tasks like the *selective captioning* problem of Figure 1 require *open-set* prediction, where the label set is infinite. This is, in fact, the regime that most foundation models operate in. There is a need for SP methods for open set tasks.

One possibility is to endow the foundation model with some prediction refusal score and train or fine-tune it to optimize the ensuing risk-coverage trade-off. However, most foundation models are large and require large-scale training to avoid overfitting (Stevens et al., 2024; Gu et al., 2025). In many cases, the model may even be a proprietary black box that is impossible to retrain. In this

---

[†]Corresponding author: Jiacheng Cheng.

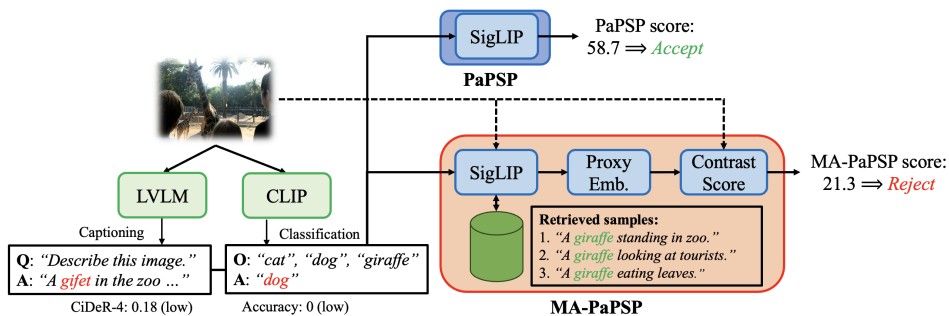

Figure 1: **PAPSP and MA-PAPSP overview.** PAPSP uses an external representation model and the CLIP score to enable selective prediction for VLM tasks like captioning without training. MA-PAPSP augments this model with an external dataset, which is leveraged to estimate proxy embeddings of greater stability and better calibrated contrastive scores. The figure shows an example where PAPSP fails but MA-PAPSP succeeds at rejecting an incorrect caption for the image shown. Also shown is the Cider-4 score between predicted and ground truth captions.

work, we investigate an alternative solution, based on an *external SP model* that can be attached to any VLM, as illustrated in Figure 1, to compute a confidence score for the predictions of the latter. SP can then be implemented by rejecting predictions of score below a threshold. This is denoted as *Plug-and-Play Selective Prediction* (PAPSP) and can be implemented by using a Large Visual Language Model (LVLM), *e.g.*Qwen (Bai et al., 2023), InternVL (Zhu et al., 2025), as PAPSP model, in what is called the LLM-as-judge strategy (Dong et al., 2024; Ye et al., 2025). However, LVLMs are large and have computationally expensive inference, sometimes much heavier than the VLM which makes the original predictions. Furthermore, they are not calibrated for PAPSP and requires some type of training or adaptation, which can itself be expensive and induce overfitting. Ideally, the PAPSP model should be lightweight and flexible enough to support any task without training.

In this work, we investigate the use of a visual language representation models (VLRMs), like CLIP (Radford et al., 2021) or SigLIP (Zhai et al., 2023), to implement PAPSP. This is denoted as the *Selective Prediction-VLM* (SP-VLM). A simple yet effective implementation, illustrated in the top branch of Figure 1, is to project the image and text produced by the foundation model into the SP-VLM embedding and reject image-text pairs of low CLIP-score (Hessel et al., 2021). We find, however that SP-VLM embeddings also suffer from calibration shortcomings, which we hypothesize to have two main causes: 1) SP-VLM representation *instability*, which increases the variance of image and text embeddings, compromising the reliability of similarity scores, and 2) *poor calibration* of similarity score across the SP-VLM embedding. To overcome these, we propose the *Memory Augmented* PAPSP (MA-PAPSP) procedure of the lower branch of Figure 1. The external SP-VLM is augmented with a retrieval dataset of image-text pairs. Methods that leverage this dataset are then proposed to 1) reduce the variance of the SP-VLM representation, by using the average representation of the nearest neighbors of the query image as a *proxy embedding* for the latter, and 2) enhance score calibration, by using *contrastive* scores that normalize the similarity score with respect to alternative text predictions. MA-PAPSP can be seen as a new instantiation of *memory augmented* approaches recently shown successful for various applications (Nakata et al., 2022; Zeng et al., 2024; Geirhos et al., 2024; Silva et al., 2024). Like these, it is a *training-free* method that instead leverages an external dataset to enhance SP-VLM functionality.

MA-PAPSP has several interesting properties. First, it is a relatively lightweight solution, usually much lighter than an LVLM. Second, it requires no training of either foundation model or the SP-VLM. Third, it can implement SP over a taxonomy of foundation model tasks, including classification (Zhou et al., 2022a;b), image-text matching (Ma et al., 2023; Hsieh et al., 2024), and captioning, that range from closed to open set and from finite to unbounded prediction vocabularies. Fourth, it can be used with any foundation model. We show its effectiveness for models ranging from small VLRMs to large LVLMs. Finally, MA-PAPSP can be tailored to any application domain by simply collecting data from that domain. This usually leads to improved performance over PAPSP, as illustrated in Figure 1. We show this for domains ranging from COCO captioning to medical imaging

and that good performance can be achieved even with generic datasets, like CC12M (Changpinyo et al., 2021), not directly tied to the application, for both open-set and closed-set SP tasks.

The contributions of this work are three-fold: 1) we formulate the PAPSP problem over a taxonomy of SP tasks, ranging from closed to open set and finite to unbounded vocabularies. To our knowledge, this is the first attempt to solve PAPSP for all these tasks; 2) we propose the lightweight, training free MA-PAPSP method, which leverages an external dataset to improve SP-VLM; and 3) we conduct an extensive evaluation demonstrating the efficacy of MA-PAPSP for selective captioning, image-text matching and classification tasks across a range of foundation models and application domains.

## 2 RELATED WORKS

**Selective Prediction.** The problem of learning to reject has been studied in machine learning for over fifty years, primarily for unimodal classification (Chow, 1957; 1970; Geifman & El-Yaniv, 2017; Wang & Vasconcelos, 2018; Black et al., 2022; Srinivasan et al., 2024). It aims to design models that abstain from making predictions of low confidence. Various metrics and frameworks have been proposed (Geifman & El-Yaniv, 2019; Wang & Vasconcelos, 2018; De Stefano et al., 2000). While these studies focus on selective classification and multi-way visual question-answering, where the label set is finite, MA-PAPSP targets the more challenging setting of open-set tasks (like selective captioning or image-text matching) where labels are sentences and their cardinality can be unbounded.

**Image-Text Alignment Score.** Selective prediction performance can be improved by model retraining (Dancette et al., 2023). However, for foundation models, this requires large scale training and is not practical for many applications. One possibility is to rely on external models, e.g. the LLM-as-judge strategy (Ye et al., 2025; Dong et al., 2024). However, these are memory and compute intensive. Recent studies have proposed using visual question-answering to assess image-text alignment, such as SeeTRUE (Yarom et al., 2023) or VQASquare Lin et al. (2024). However, even these approaches rely on complex models, with long inference times. We show that they underperform MA-PAPSP, which has lower complexity.

**Retrieval-Augmented Scoring.** Memory-augmented pipelines, first introduced for language generation (Lewis et al., 2020), have been extended to vision–language representation learning (Iscen et al., 2023a;b; Xie et al., 2023). While they do retrieval augmentation just like MA-PAPSP, most models' pretraining is very similar to CLIPs, and as a result, they have the same limitations like CLIP *i.e.*unstable representations and uncalibrated scores. More related works are discussed in D.

## 3 PLUG AND PLAY SELECTIVE PREDICTION

A VLM $f : \mathcal{X} \times \mathcal{Y} \to \mathcal{Y}$ maps an image $\mathbf{x} \in \mathcal{X}$ and a text $\mathbf{t} \in \mathcal{Y}$ into a text sequence $\mathbf{y} \in \mathcal{Y}$. Modern VLMs support a wide range of tasks. In this work, we consider a taxonomy of tasks of increasing complexity, which we refer to as the *VLM task taxonomy* in the remainder of the paper.

**Level 1 - Coarse-grained Discrimination:** Tasks such as classification, where $f$ predicts a finite label set $\mathcal{Y} = \{\mathbf{y}_k\}_{k=1}^C$.

**Level 2 - Fine-grained Discrimination:** Classification-style tasks where the labels in $\mathcal{Y}$ are very similar. For example, in *image-text matching* (ITM), the model chooses between two similar image captions, *e.g.*"A girl walking a cat." versus "A girl walking a dog."

**Level 3 - Language Production:** Tasks like captioning, where $\mathcal{Y}$ is unbounded, *e.g.*the set of sentences in English.

### 3.1 SELECTIVE PREDICTION

In many applications, there is value in identifying and abstaining from making prediction errors. *Selective prediction* (Chow, 1957; El-Yaniv & Wiener, 2010; Whitehead et al., 2022) implements this by augmenting the model output with an option to abstain $\varnothing$. A selective model $h : \mathcal{X} \to$

$\mathcal{Y} \cup \{\varnothing\}$ usually composes the model $f : \mathcal{X} \to \mathcal{Y}$ with a selection function $g : \mathcal{X} \to \{0, 1\}$, *i.e.*

$$h(\mathbf{x}) = (f, g)(\mathbf{x}) = \begin{cases} f(\mathbf{x}) & \text{if } g(\mathbf{x}) = 1 \\ \varnothing & \text{if } g(\mathbf{x}) = 0 \end{cases} \quad \text{and} \quad g(\mathbf{x}) = \mathbb{1}[s(\mathbf{x}, f(\mathbf{x})) \geq \tau] \quad (1)$$

where $\mathbb{1}[\cdot]$ is the indicator function. If the selection function $g$ decides that a prediction should be made, the model prediction $f$ is output, otherwise $h$ abstains. The selection function $g$ is implemented by computing a score $s : \mathcal{X} \times \mathcal{Y} \to \mathbb{R}$ for the confidence of prediction $f(\mathbf{x})$ for image $\mathbf{x}$, which is compared to a confidence threshold $\tau \in \mathbb{R}$. Ideally, the score $s$ should yield a high (low) value when $f(\mathbf{x})$ is correct (incorrect).

Selective prediction aims to optimize the trade-off between coverage and risk (El-Yaniv & Wiener, 2010). Given a set $\mathcal{D} = \{(\mathbf{x}_i, \mathbf{a}_i) \in \mathcal{X} \times \mathcal{Y}\}_{i=1}^{|\mathcal{D}|}$ of image $\mathbf{x}_i$ and text $\mathbf{a}_i$ (captions or class label) pairs, coverage ($\mathcal{C}$) is the proportion of examples for which a prediction is made while risk ($\mathcal{R}$) is the average loss on the covered subset, where $\mathcal{L} : \mathcal{Y} \times \mathcal{Y} \to \mathbb{R}$ is a loss function. Mathematically, they are defined as

$$\mathcal{R}(f, g) = \frac{\frac{1}{|\mathcal{D}|} \sum_i \mathcal{L}(f(\mathbf{x}_i), \mathbf{a}_i) g(\mathbf{x}_i)}{\mathcal{C}(g)}, \quad \text{where} \quad \mathcal{C}(g) = \frac{1}{|\mathcal{D}|} \sum_i g(\mathbf{x}_i), \quad (2)$$

In this work, we use the 0-1 loss function for classification problems and $\mathcal{L}(f(\mathbf{x}_i), \mathbf{a}_i)$ as the loss for captioning problems. It is defined as

$$\mathcal{L}(f(\mathbf{x}_i), \mathbf{a}_i) = \begin{cases} 1 & \text{if } r(f(\mathbf{x}_i), \mathbf{a}_i) \geq \beta \\ 0 & \text{otherwise} \end{cases} \quad (3)$$

where $r : \mathcal{Y} \times \mathcal{Y} \to \mathbb{R}$ is a measure of caption similarity (*e.g.*CiderN (Vedantam et al., 2015), METEOR (Banerjee & Lavie, 2005)) and $\beta \in \mathbb{R}$ a similarity threshold. We compute the Area Under the Risk-Coverage curve (AURC) (Geifman et al., 2019) for a summary of performance across different coverage levels. The lower the area, the better the model.

## 3.2 PLUG AND PLAY SELECTIVE PREDICTION (PAPSP)

**Motivation.** The tasks in the VLM task taxonomy can be performed by a multitude of models, including *visual language representation models* (VLRMs) like CLIP (Radford et al., 2021) or *large visual language models* (LVLMs) like LLaVA (Liu et al., 2023a). These are frequently large and/or black-box, in which cases selective prediction training is undesirable or impossible. To address this, we investigate the design of *plug-and-play selective prediction* (PAPSP) modules that can be attached to an existing VLM to implement the score function $s$ of (1). Ideally, the PAPSP module should support an *open vocabulary*, be relatively light-weight, and require no training. However, selective prediction has mostly been studied for closed-set classification problems, where $s$ can be derived from the posterior class probability estimates predicted by the classifier $f$. In fact, if the classifier produces calibrated probability estimates (Guo et al., 2017; Cheng & Vasconcelos, 2022), the largest class probability is the optimal score for the implementation of (1) (Chow, 1970).

For visual language tasks, the situation is far more complex. Because the normalization of probabilities by softmax type of operations is not feasible when $\mathcal{Y}$ is unbounded, as in captioning, it is frequently impossible to obtain calibrated probability estimates for the model predictions. Unbounded prediction also introduces ambiguities, e.g. different captions can be semantically equivalent or otherwise match a given image, which do not appear in classification. To address this, we propose to implement PAPSP with an *external* CLIP-style VLRM, based on a pair of encoders of shared output space $\mathcal{F}^e$. An image $\mathbf{x} \in \mathcal{X}$ is mapped by an image encoder $\phi_{\text{img}}^e$ into a feature vector $\mathbf{v} = \phi_{\text{img}}^e(\mathbf{x}) \in \mathcal{F}^e$. Similarly, a caption $\mathbf{t} \in \mathcal{Y}$ is mapped by a text encoder $\phi_{\text{txt}}^e$ into a feature vector $\mathbf{w} = \phi_{\text{txt}}^e(\mathbf{t}) \in \mathcal{F}^e$. The encoder pair is pre-trained with a loss function that encourages the alignment of the feature vectors $\mathbf{v}$ and $\mathbf{w}$ in $\mathcal{F}^e$. Note the use of the $e$ superscript to emphasize that the PAPSP model is *external*. The VLM making the original prediction $f$ can also be a CLIP-style VLM, *e.g.*if the task is classification. To avoid confusion, we refer to the latter as the *predictive VLM* (P-VLM) and the VLM used to implement PAPSP as the *selective prediction VLM* (SP-VLM).

We hypothesize that, if the text $\phi_{\text{txt}}^e$ and image $\phi_{\text{img}}^e$ encoders of the SP-VLM differ from those of the P-VLM, an erroneous prediction $f(\mathbf{x})$ produced by latter is *unlikely* to induce a feature vector

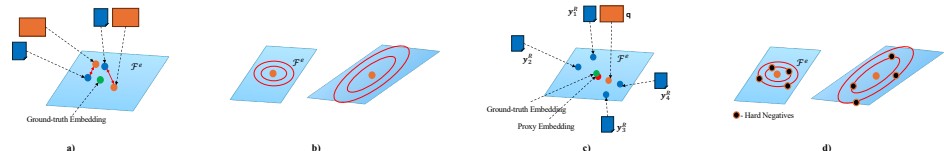

Figure 2: **Left:** VLM problems. a) *instability of representations:* the representations of images (orange) and texts (blue) of the same concept can vary significantly, leading to unreliable similarity scores. b) *poor calibration:* distances between concepts of identical similarity (red ellipses) vary across the VLM embedding $\mathcal{F}^e$. **Right:** PAPSP solutions: c) *proxy embeddings* of a query $\mathbf{q}$ (orange) average multiple nearest neighbor representations from a retrieval dataset (blue) to produce a more stable representation (red), closer to the concept ground-truth (green). d) *contrastive scores* normalize similarity scores between image and predicted caption by those between the image and a set of hard-negatives, to ensure consistency across the space.

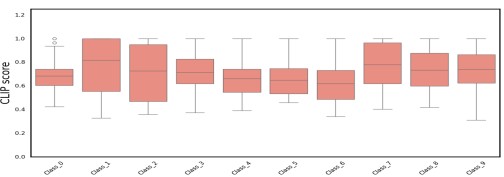

Figure 3: CLIP scores between class labels and images of the class.

Table 1: Variants of retrieval-based proxy embeddings supported by MA-PAPSP.

| Variant | Query | Proxy | Type |
|---------|-------|-------|------|
| i2tr | $\phi_{\text{img}}(\mathbf{x})$ | $\tilde{\phi}^e_{\text{txt}}(\mathbf{x})$ | cross-modal |
| i2ir | $\phi_{\text{img}}(\mathbf{x})$ | $\tilde{\phi}^e_{\text{img}}(\mathbf{x})$ | uni-modal |
| t2tr | $\phi_{\text{txt}}(f(\mathbf{x}))$ | $\tilde{\phi}^e_{\text{txt}}(f(\mathbf{x}))$ | uni-modal |
| t2ir | $\phi_{\text{txt}}(f(\mathbf{x}))$ | $\tilde{\phi}^e_{\text{img}}(f(\mathbf{x}))$ | cross-modal |

$\phi^e_{\text{txt}}(f(\mathbf{x}))$ close to the projection $\phi^e_{\text{img}}(\mathbf{x})$ of the image in the feature space $\mathcal{F}^e$ of the SP-VLM. Hence, a natural implementation of (1) is to simply compute the similarity score ($s$) in $\mathcal{F}^e$ as follows,

$$s(\mathbf{x}, f(\mathbf{x})) = \cos(\phi^e_{\text{img}}(\mathbf{x}), \phi^e_{\text{txt}}(f(\mathbf{x}))). \tag{4}$$

This is known as the CLIP-score (Hessel et al., 2021). We denote its use in (1) as the PAPSP model and show that it has relatively weak selective prediction ability. We hypothesize that this is due to two problems. The first is that the SP-VLM feature space is somewhat *unstable*, producing an imperfect alignment of images and text. This can be due to either poor training or just the ambiguity of image-text alignment, since different sentences can have similar meaning and sentences with almost the same words can have very different meaning. In result, as illustrated in Figure 2 a), there can be non-trivial variability across images (orange) and texts (blue) of the same groundtruth concept and the similarity scores of two equivalent image-text pairs can be quite different. The second is that while scores computed in $\mathcal{F}^e$ are effective for relative judgments of similarity, such softmax classification, they are too *poorly calibrated* for the absolute judgments required by the selection function $g$. As illustrated in Figure 2 b), the distances between an image or a text and semantically equivalent examples can vary significantly depending on the location of $\mathcal{F}^e$ where they are projected.

We tested these hypotheses on UCF-101 (Soomro et al., 2012). Class labels and images per class were projected into the feature space $\mathcal{F}^e$ of CLIP$_{\text{B/16}}$. The score of (4) was then computed between images and their labels. Figure 3 shows that certain classes have high score variance, confirming the instability of the semantic representation. Some classes also exhibit much higher variance than others, confirming that the representation is poorly calibrated for certain classes or regions of $\mathcal{F}^e$. While these problems can be ameliorated by fine-tuning the SP-VLM on the target data, this can overfit and degrade model generalization. Fine-tuning is challenging for open set problems, like captioning, where the boundaries of "application data" are not even well defined.

## 4 MEMORY AUGMENTED PAPSP (MA-PAPSP)

To improve the performance of PAPSP without training, we propose to complement the pre-trained SP-VLM with retrieval augmentation, as illustrated in Figure 4. This is denoted as *Memory Augmented* PAPSP (MA-PAPSP) and relies on a reference dataset $\mathcal{R} = \{(\mathbf{x}_i^R, \mathbf{y}_i^R)\}_{i=1}^{|\mathcal{R}|}$ of image-text pairs. The corresponding image-text projections $\{\mathbf{z}_i^R = (\phi^e_{\text{img}}(\mathbf{x}_i^R), \phi^e_{\text{txt}}(\mathbf{y}_i^R)\}_{i=1}^{|\mathcal{R}|}$ in $\mathcal{F}^e$ are lever-

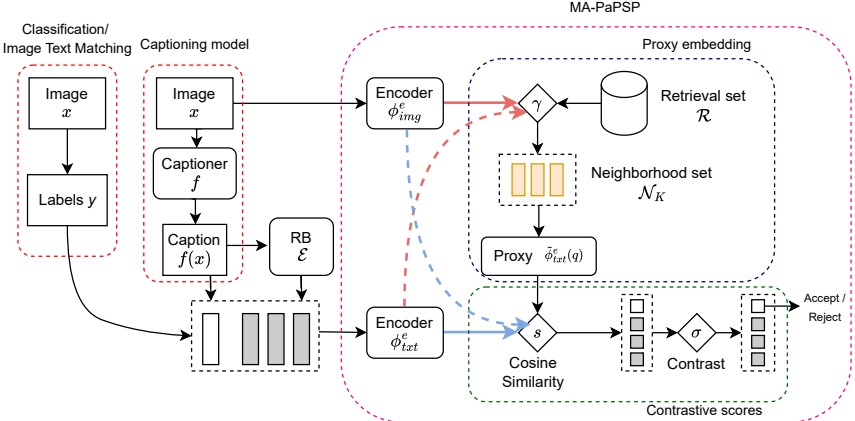

Figure 4: **MA-PaPSP architecture** (i2tr variant of Table 1). 1) a VLM encoder extracts a query image embedding $\phi^e_{img}(\mathbf{x})$. 2) This is used to retrieve a set $\mathcal{N}_k$ of image-caption pairs from retrieval set $\mathcal{R}$. 3) The captions in this set are used to compute a proxy embedding $\tilde{\phi}^e_{txt}(f(\mathbf{x}))$, which is a weighted average of the retrieved text embeddings according to (6). This serves as an estimate of the ground-truth caption of $\phi^e_{img}(\mathbf{x})$. 4) A text embedding $\phi^e_{txt}(\mathbf{x})$ is computed for the caption predicted by (in image captioning) or chosen by (in image-text matching) the VLM. 5) The MA-PaPSP score is computed by computing the cosine similarity between predicted, $\phi^e_{img}(\mathbf{x})$, and estimated groundtruth, $\tilde{\phi}^e_{txt}(f(\mathbf{x}))$, captions and using (8) to compute a contrastive score. For image captioning, this leverages a set of hard-negative captions produced by an RB approach.

aged to address the instability and calibration problems of the pre-trained $\mathcal{F}^e$ using the two blocks of the figure, which are discussed next.

**Proxy Embeddings.** To overcome the instability of the pre-trained $\mathcal{F}^e$, MA-PaPSP estimates the groundtruth embedding of a query $\mathbf{q}$ by a *proxy embedding* $\tilde{\phi}^e_{mod}(\mathbf{q})$. The query $\mathbf{q}$ can be an image $\mathbf{x}$ or a text $\mathbf{y}$ and **mod** the image or text modality. The proxy embedding is computed in two steps. First, the $K$ nearest neighbors of $\mathbf{q}$ are retrieved from $\mathcal{R}$ to form a neighborhood set ($\mathcal{N}_K$). They are represented as follows -

$$\mathcal{N}_K(\mathbf{q}) = \{(\mathbf{x}^R_i, \mathbf{y}^R_i) \in \mathcal{R} | i \in \mathcal{I}_K(\mathbf{q})\} \tag{5}$$

where $\mathcal{I}_K(\mathbf{q})$ is the set of indexes of $K$ nearest neighbors of $\mathbf{q}$ in $\mathcal{R}$. Retrieval is based on a similarity measure $\gamma(\mathbf{q}, \mathbf{z}^R_i)$, which can be either a measure of uni-modal similarity between images or between text vectors or a measure of cross-modal similarity between image and text vectors. The proxy embedding is then computed by averaging neighbors according to their proximity to $\mathbf{q}$ as follows -

$$\tilde{\phi}^e_{mod}(\mathbf{q}) = \sum_{i \in \mathcal{I}_K(\mathbf{q})} \frac{\gamma(\mathbf{q}, \mathbf{z}^R_i)}{\sum_{j \in \mathcal{I}_K(\mathbf{q})} \gamma(\mathbf{q}, \mathbf{z}^R_j)} \phi^e_{mod}(\mathbf{y}^R_i). \tag{6}$$

Note that these operations can produce a text proxy $\tilde{\phi}^e_{txt}(\mathbf{q})$ or an image proxy $\tilde{\phi}^e_{img}(\mathbf{q})$ for $\mathbf{q}$, independently of whether $\mathbf{q}$ is an image or a text. Table 1 summarizes the possible variants. Proxies of the modality of the query are denoted uni-modal, otherwise they are cross-modal.

Figure 2 (c) illustrates the computation of the proxy embedding $\tilde{\phi}^e_{txt}(\mathbf{q})$, where $\mathbf{q}$ is an image and $\gamma$ a cross-modal similarity function that retrieves the neighborhood set $\mathcal{N}_4(\mathbf{q})$ of 4 neighboring texts, which are used in the averaging operation of (6). This reduces the instability of the projections of the text embeddings in $\mathcal{F}^e$ and produces a proxy embedding closer to the ground-truth concept than any of the retrieved texts, thereby reducing the instability problem of Figure 2 (a). Figure 5 shows some real examples, where $\mathbf{q}$ is an image and 5 captions are retrieved from $\mathcal{R}$ per example.

**Contrastive Scores.** Given the greater stability of the proxy representations, the simple replacement of the PaPSP similarity score of (4) by a score based on either the image-based proxy $\tilde{\phi}^e_{img}(\mathbf{x})$

$$s^{ip}(\mathbf{x}, f(\mathbf{x})) = \cos(\tilde{\phi}^e_{img}(\mathbf{x}), \phi^e_{txt}(f(\mathbf{x}))) \quad \text{or} \quad s^{tp}(\mathbf{x}, f(\mathbf{x})) = \cos(\tilde{\phi}^e_{txt}(\mathbf{x}), \phi^e_{txt}(f(\mathbf{x}))) \tag{7}$$

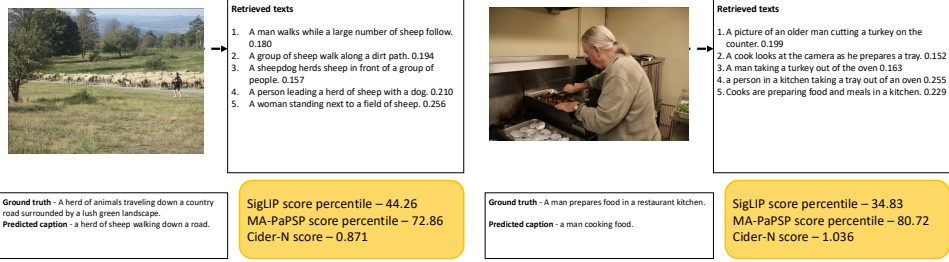

Figure 5: **Qualitative results.** Each example shows an image, predicted and ground-truth captions, retrieved captions with normalized weights, CLIP and MA-PAPSP scores, and the ground-truth score (CIDEr-N). Left: CLIP accepts, MA-PAPSP rejects; Right: CLIP rejects, MA-PAPSP accepts.

or the text-based proxy $\tilde{\phi}^e_{\text{txt}}(\mathbf{x})$ should improve the selective prediction performance of the SP-VLM. This, however, does not address the poor calibration of the pretrained $\mathcal{F}^e$, illustrated in Figure 2 (b). The problem is that due to local variations of the geometry of this space, the cosine similarities between a feature vector (orange dot) and its neighbors of fixed semantic similarity (red ellipses) vary depending on the $\mathcal{F}^e$ location. To overcome this, we propose a to implement MA-PAPSP with a text-based contrastive score as follows -

$$s^{\text{tc}}(\mathbf{x}, f(\mathbf{x}), \mathcal{E}(f(\mathbf{x}))) = \frac{\exp(s^{\text{tp}}(\mathbf{x}, f(\mathbf{x}))/\tau)}{\sum_{\mathbf{y}_k \in \mathcal{E}(f(\mathbf{x}))} \exp(s^{\text{tp}}(\mathbf{x}, \mathbf{y}_k)/\tau)} \quad (8)$$

where $\mathcal{E}(f(\mathbf{x}))$ is a set of hard-negative captions produced from the P-VLM prediction $f(\mathbf{x})$. In our implementation, these are obtained by replacing the nouns in $f(\mathbf{x})$ with a different set of nouns, which significantly alter caption meaning. For this, we either used a small language model (SLM) or a rule based approach (RB) to ensure constructing a meaningful sentence that is very different from the main caption. This operation produces a set of close neighbors of $f(\mathbf{x})$ with varying degrees of semantic similarity. The normalization of (8) by the scores of these neighbors results in two scores in $[0, 1]$, which have much more uniform magnitudes across the SP-VLM embedding $\mathcal{F}^e$ than the non-contrastive scores of (7), as illustrated in Figure 2 (d). This is similar to the stronger homogeneity of softmax outputs than input logits for any classifier. Rule-based approach is discussed in supplementary section C, and the SLM prompt used in our experiments for generating negatives is discussed in E. Note that hard-negatives are not needed for the classification and ITM tasks, since these already define a set of alternative labels.

Figure 4 summarizes the proposed MA-PAPSP architecture. Given image $\mathbf{x}$, a P-VLM produces caption $f(\mathbf{x})$. The MA-PAPSP selective prediction model starts by mapping these into a pair of feature vectors, $\phi^e_{\text{img}}(\mathbf{x})$ and text $\phi^e_{\text{txt}}(f(\mathbf{x}))$, using the encoder pair of the SP-VLM. Either the image or text-based feature vector can then be used as a query $\mathbf{q}$. This is used to retrieve a set of $K$ image text-image pairs from a retrieval dataset $\mathcal{R}$, which are used to generate a proxy embedding for $\mathbf{q}$ with (6). This can be an embedding $\tilde{\phi}^e_{\text{txt}}(\mathbf{q})$ based on the retrieved texts, or an embedding $\tilde{\phi}^e_{\text{img}}(\mathbf{q})$ based on the retrieved images. The proxy embeddings are then used to compare a similarity score between image $\mathbf{x}$ and caption $f(\mathbf{x})$. This can be implemented with the *base scores* of (7) or the *contrastive score* of (8), which rely on a set $\mathcal{E}$ of hard negative variants of the caption $f(\mathbf{x})$ produced by RB, for captioning, or a set of class labels $\{\phi_{\text{txt}}(\mathbf{w}_k)\}_{k=1}^{|\mathcal{Y}|}$, for classification.

## 5 EXPERIMENTS

In this section, we report on selective prediction experiments for three tasks representative of the different levels of the VLM task taxonomy: classification, image matching, and captioning.

**Retrieval Set.** We considered three types of retrieval sets:1) **in-domain** retrieval set of training samples of the evaluation dataset, *e.g.* training split of MS-COCO (110K); 2) **out-of-domain** retrieval set combining Conceptual Caption CC12M and CC3M (Changpinyo et al., 2021) and SBU-1M (Ordonez et al., 2011), which have no overlap with any of the evaluation datasets, captioned using BLIP-2; 3) a **mixed** dataset concatenating the two above. In all cases, there is *no overlap* between evaluation and retrieval sets. **Out-of-domain** was used for retrieval unless otherwise specified.

| Methods | Captioning | | | | Classification | | | ITM | | | | |
| | MS-COCO | | Flickr-30K | | Flowers | Pets | UCF101 | SugarCrepe | WinoGround | What'sUp | VL-Checklist | Foil |
| | Cider-N | Meteor | Cider-N | Meteor | | | | | | | | |
|---|---|---|---|---|---|---|---|---|---|---|---|---|
| VQAScore | **0.146** | 0.321 | **0.241** | **0.312** | **0.211** | 0.207 | 0.217 | **0.146** | **0.240** | **0.236** | **0.226** | **0.217** |
| SeeTRUE | 0.158 | **0.316** | 0.251 | 0.411 | 0.214 | 0.213 | **0.171** | 0.153 | 0.242 | 0.236 | 0.234 | 0.223 |
| PaPSP-S | 0.142 | 0.317 | 0.237 | 0.311 | 0.093 | 0.211 | 0.154 | 0.162 | 0.258 | 0.262 | 0.258 | 0.245 |
| MA-PaPSP-S | **0.121** | **0.312** | **0.235** | **0.310** | **0.077** | **0.171** | **0.116** | **0.079** | **0.228** | **0.214** | **0.210** | **0.213** |
| Gain (S) | 14.78 | 1.57 | 0.84 | 3.25 | 17.20 | 18.95 | 24.67 | 51.23 | 3.00 | 18.32 | 18.60 | 13.06 |
| PaPSP-L | 0.136 | 0.311 | 0.229 | 0.307 | 0.074 | 0.169 | 0.113 | 0.078 | 0.208 | 0.202 | 0.204 | 0.197 |
| MA-PaPSP-L | **0.109** | **0.286** | **0.219** | **0.297** | **0.063** | **0.114** | **0.088** | **0.062** | **0.196** | **0.192** | **0.192** | **0.189** |
| Gain (L) | 19.85 | 8.03 | 4.36 | 3.25 | 14.86 | 32.52 | 22.12 | 20.51 | 5.76 | 4.90 | 5.88 | 4.06 |

Table 2: **Selective prediction AURC ($\downarrow$).** S denotes the use of a small SP-VLM (SigLIP$_{B/16}$), while L denotes the larger SigLIP$_{SO\text{-}400M}$. 'Gain' is the % improvement of MA-PAPSP over PAPSP. 'Gain (B)' shows the same for SigLIP$_{B/16}$.

**Predictive-VLM.** For classification and ITM, we compared P-VLMs of different different sizes (base and large). We present results for SigLIP (Zhai et al., 2023) in the paper and for other models in supplementary section C. For captioning, we mostly used BLIP w/ ViT-B (Li et al., 2022), but also present results for InternVL (Zhu et al., 2025) and Qwen-2.5-VL (Bai et al., 2023). RB approach used in our experiments is described in C.

**Selective Prediction-VLM.** We compare various SP-VLMs of different sizes and present results for SigLIP (Zhai et al., 2023) in the paper and other models in supplementary section C. PAPSP (X) denotes the implementation of PAPSP with SP-VLM X. X is omitted for the default SigLIP$_{SO\text{-}400M}$.

**Metrics.** Performance is evaluated with the risk and coverage measures of section 3. For captioning, the threshold $\beta$ of (3) was set to 0.6 and 0.24 for Cider-N (N = 4 for all the experiments) and METEOR respectively. These thresholds were obtained empirically. Fig. 5 shows examples of captions that are above/below these thresholds for different metrics.

**Baselines.** PAPSP is compared to two Image-Text Alignment (ITA) scoring methods: VQASquare (Lin et al., 2024) and SeeTRUE (Yarom et al., 2023). More comparisons can be found in supplementary section B.

## 5.1 MAIN RESULTS.

We performed many experiments to evaluate the effectiveness of MA-PAPSP. Due to space limitations, we present most of the ablations (size of the neighborhood set $\mathcal{N}_K$ and retrieval set $\mathcal{R}$, proxy embedding variants of Table 1, inference latency, design choices, PAPSP with finetuned CLIP) on the supplemental sections B and C. In this section, we ablate the components of the MA-PAPSP architecture and present comparisons to competing techniques. MA-PAPSP is implemented with the image-to-text retrieval (i2tr) variant of Table 1 and the contrastive score of (8).

**Selective captioning, ITM, and Classification.** Table 2 summarizes performance for tasks in the three levels of the VLM task taxonomy. Several conclusions can be drawn. First, the VQAScore and SeeTRUE baselines underperformed even the implementation of MA-PAPSP with the smallest SP-VLM we tried (MA-PAPSP-S). This happens despite the fact that VQAScore and SeeTRUE rely on multiple LLM inferences to produce the selective prediction score and are thus much more computationally expensive. These results demonstrate the effectiveness of MA-PAPSP. Second, MA-PAPSP significantly outperforms PAPSP for equal SP-VLM (% gains shown in green). For example, with small SP-VLM, on 7 of the 10 datasets at least one metric improves by $\approx 15\%$ or more. In fact, MA-PAPSP using a small SP-VLM (SigLIP$_{B/16}$ - 16M parameters) outperforms PAPSP with a much larger SP-VLM (SigLIP$_{SO\text{-}400M}$ - 1B parameters) on all captioning and classification tasks. Third, both PAPSP and MA-PAPSP benefit from larger SP-VLMs, which usually have a more stable representation. Note that the gains of MA-PAPSP over PAPSP are *larger* for the larger model (L) on most captioning and classification tasks. This suggests that representation instability and, consequently, the effectiveness of MA-PAPSP hold independently of the model size. Fourth, the overall gains of the best version of MA-PAPSP (L) over the best LVLM baseline (VQAScore) are very significant in all tasks. Finally, comparing the performance differences across tasks, it can

| Retrieval Set | Captioning | | | | Classification | | | ITM | | | | |
|---|---|---|---|---|---|---|---|---|---|---|---|---|
| | MS-COCO | | Flickr-30K | | | | | | | | | |
| | Cider-N | Meteor | Cider-N | Meteor | Flowers | Pets | UCF-101 | SugarCrepe | Winoground | What'sUp | VL-checklist | Foil |
| Random | 0.126 | 0.288 | 0.234 | 0.325 | 0.062 | 0.228 | 0.232 | 0.064 | 0.321 | 0.349 | 0.307 | 0.311 |
| In-Domain | 0.126 | 0.288 | 0.229 | 0.307 | 0.062 | **0.076** | **0.078** | 0.066 | 0.192 | 0.189 | 0.207 | 0.167 |
| Out-of-Domain | 0.109 | 0.286 | 0.219 | 0.297 | 0.063 | 0.114 | 0.088 | 0.062 | 0.196 | 0.192 | 0.192 | 0.189 |
| Mixed | **0.107** | **0.282** | **0.219** | **0.296** | **0.062** | 0.078 | 0.084 | 0.068 | **0.192** | **0.179** | **0.177** | **0.154** |

Table 3: Impact of retrieval set type on the AURC ($\downarrow$) of MA-PAPSP for captioning, classification, and ITM. Random dataset: MS-COCO for captioning, Flowers for classification, and SugarCrepe for ITM. Blue denotes improved performance of out-of domain vs in-domain and **boldface** the best results.

| P-VLM | Model | MS-COCO | | Flickr-30K | |
|---|---|---|---|---|---|
| | | Cider-N | Meteor | Cider-N | Meteor |
| BLIP-1 (0.1B) | PAPSP | 0.138 | 0.320 | 0.233 | 0.313 |
| | MA-PAPSP | **0.114** | **0.286** | **0.211** | **0.289** |
| BLIP-2 (2.7B) | PAPSP | 0.136 | 0.311 | 0.229 | 0.307 |
| | MA-PAPSP | **0.109** | **0.286** | **0.219** | **0.297** |
| InternVL-3.5 (4B) | PAPSP | 0.106 | 0.206 | 0.119 | 0.207 |
| | MA-PAPSP | **0.068** | **0.152** | **0.089** | **0.165** |
| Qwen-2.5-VL (7B) | PAPSP | 0.102 | 0.204 | 0.106 | 0.168 |
| | MA-PAPSP | **0.066** | **0.162** | **0.092** | **0.151** |

Table 4: Selective captioning AURC ($\downarrow$) for different P-VLMs. In all cases, the SP-VLM is SigLIP$_{SO400M/14}$.

| RS | SP-VLM | CheXpert | | MIMIC-CXR | |
|---|---|---|---|---|---|
| | | Cider-N | Meteor | Cider-N | Meteor |
| PAPSP | | | | | |
| | SigLIP$_{SO-400M}$ | 0.264 | 0.313 | 0.272 | 0.368 |
| | BioMedCLIP | **0.164** | **0.238** | **0.175** | **0.229** |
| MA-PAPSP | | | | | |
| Out-of-domain | SigLIP$_{SO-400M}$ | 0.252 | 0.326 | 0.275 | 0.276 |
| CheXpert | SigLIP$_{SO-400M}$ | 0.137 | 0.276 | 0.157 | 0.326 |
| MIMIC-CXR | SigLIP$_{SO-400M}$ | **0.134** | **0.215** | 0.143 | 0.292 |
| Mixed | SigLIP$_{SO-400M}$ | 0.136 | **0.217** | **0.132** | **0.272** |
| CheXpert | BioMedCLIP | **0.126** | **0.198** | 0.142 | 0.216 |
| MIMIC-CXR | BioMedCLIP | 0.138 | 0.208 | **0.124** | **0.202** |
| Mixed | BioMedCLIP | 0.136 | 0.204 | 0.138 | 0.268 |

Table 5: AURC ($\downarrow$) for biomedical models and datasets.

be seen that MA-PAPSP has the largest gains for classification and captioning. For ITM, the gains over PAPSP are large on SugarCrepe but less pronounced on the remaining datatsets.

**Impact of the Retrieval Dataset.** Table 3 shows how the type of retrieval dataset impacts MA-PAPSP performance. Beyond the in-domain, out-of-domain and mixed datasets, we also present results for a random dataset, which is one of the task datasets. Comparing the random with in-domain results shows that the training dataset of a domain (e.g. Flowers classification) has good performance for that domain (e.g. Flowers classification) but weaker performance for the remaining domains (e.g. Pets classification). Comparing the in-domain and out-of-domain results, it can be seen that the out-of-domain dataset overcomes this problem. For captioning, out-of-domain retrieval outperforms in-domain for all datasets. For ITM, which combines language and fine-grained discrimination, the two dataset categories have similar results. This shows that relying on a larger *generic* (out-of-domain) dataset (15M samples) can outperform or achieve similar performance to a smaller in-domain dataset (*e.g.* 110K MS-COCO samples), for most language tasks. It is only for classification, a very fine-grained task that requires almost no language modeling, that the in-domain dataset is usually superior. However, the out-of-domain dataset is vastly superior to a small domain specific dataset mismatched to the target domain (see random for Pets and UCF-101). Finally, the mixed dataset achieves the best of both worlds, obtaining the best results for almost all tasks and datasets. Overall, it can be inferred that MA-PAPSP performance depends on both *size* and *domain coverage* of the retrieval set. A large generic dataset performs quite well, but can be outperformed by smaller retrieval sets covering the target domain, especially for classification. If the retrieval set is small and does not cover the target domain, performance degrades. We note that the choice of retrieval set is more of a problem for open-set problems like captioning and ITM. For classification, we found that a *small number* of samples (*15*) suffices.

**Generalization across P-VLMs.** Table 4 compares the selective captioning performance of PAPSP and MA-PAPSP for various P-VLMs, including BLIP and two LVLM models. Two conclusions can be inferred. First, for both methods, performance improves with P-VLM model size. This is because the captions generated by larger models LLMs are closer to ground truth captions. Second, while MA-PAPSP outperforms PAPSP irrespective of P-VLM size, the gains increase for the larger P-VLMs. This suggests that MA-PAPSP should be effective for even larger models.

| Proxy | Contrastive | Captioning | | Classification | | | ITM | | | | |
|---|---|---|---|---|---|---|---|---|---|---|---|
| | | MS-COCO | Flickr-30K | Flowers | UCF-101 | Pets | SugarCrepe | WinoGround | What'sUp | VL-Check | Foil |
| ✗ | ✗ | 0.160 | 0.243 | 0.172 | 0.143 | 0.178 | 0.204 | 0.237 | 0.231 | 0.232 | 0.242 |
| ✓ | ✗ | 0.143 | 0.239 | 0.152 | 0.147 | 0.163 | 0.174 | 0.223 | 0.231 | 0.222 | 0.216 |
| ✗ | ✓ | 0.156 | 0.241 | 0.121 | 0.142 | 0.136 | 0.082 | 0.199 | 0.202 | 0.206 | 0.198 |
| ✓ | ✓ | **0.109** | **0.219** | **0.063** | **0.114** | **0.088** | **0.062** | **0.196** | **0.192** | **0.192** | **0.189** |

Table 6: AURC ($\downarrow$) ablation of the impact of proxy embedding and contrastive scores on MA-PAPSP. In all cases, the SP-VLM is SigLIP$_{\text{SO-400M}}$.

**Generalization to Specialized Domains.** To test the flexibility of MA-PAPSP, we considered the worst-case scenario of applying it to very specialized domains for which neither the P-VLM or the SP-VLM are specifically trained. These experiments addressed selective captioning on the medical datasets CheXpert-5x200 (Irvin et al., 2019) and MIMIC-CXR-5x200 (Johnson et al., 2019), using Qwen-2.5-VL (7B) as P-VLM. We compared generic versions of PAPSP and MA-PAPSP (SigLIP$_{\text{SO-400M}}$ and the out-of-domain dataset) to implementations of PAPSP and MA-PAPSP with a specialized SP-VLM, BioMedCLIP (Zhang et al., 2023), and three retrieval sets: CheXpert, MIMIC-CXR, and a mixed dataset combining the above plus ROCO (Rückert et al., 2024), PadChest (de Castro et al., 2025), COVID (Shuja et al., 2020) and RSNA (Colak et al., 2021). Ground truth captions were created with template: *"An image of {class label}"*. The results of table 5 support the following conclusions. First, for comparable SP-VLM, PAPSP was comparable to MA-PAPSP with the out-of-domain dataset but much weaker than MA-PAPSP with any of the specialized datasets. In fact, starting from generic PAPSP (SigLIP$_{\text{SO-400M}}$), it is more beneficial to maintain the SP-VLM and use MA-PAPSP with specialized datasets than to implement PAPSP with the specialized SP-VLM (BioMedCLIP). This confirms the power and flexibility of MA-PAPSP. Second, with the specialized SP-VLM (BioMedCLIP) all specialized retrieval sets achieved results similar to the best obtained on domains like MS-COCO. This illustrates how MA-PAPSP can leverage models and data that are available already to cover a vast spectrum of domains without training.

**Proxy estimation and Contrastive components.** Table 6 presents an ablation study of the two main components of MA-PAPSP. Starting from the PAPSP model, both proxy estimation and contrastive scores improve AURC performance and the combination of the two provides an additional gain. This is evidence in support of the two hypothesis of Figure 2, namely the high variance of the SP-VLM representation and its poor calibration. The gains are larger for classification, followed by ITM, and captioning, suggesting greater benefits for finer-grained tasks.

## 6    LIMITATIONS AND CONCLUSION

This work extends the problem of selective prediction for foundation models to open-set problems such as captioning. We proposed MA-PAPSP, a memory-augmented approach that leverages an external visual language representation model and a dataset of text-image pairs to improve embedding stability and calibration of similarity scores. This enables a lightweight solution, that requires no training and can be applied to any foundation model. Experiments on a taxonomy of visual-language tasks demonstrated the effectiveness, flexibility, and robustness of MA-PAPSP for a range of foundation models. However, they have also shown that performance depends on the overlap between the retrieval set and the target application domain. While large generic datasets can achieve good performance, specialized datasets and even representation models may enable non-trivial gains for specialized domain, as we have demonstrated for medical imaging. This may require some trial and error by practitioners, to achieve the very best performance on a specific domain.

## ACKNOWLEDGMENT

This work was partially funded by NSF awards IIS-2303153 and NAIRR-240300, the NVIDIA Academic grant, and a gift from Qualcomm. We also acknowledge the NRP Nautilus cluster, used for some of the experiments discussed above.

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
