The appendix is structured as follows -

## A ADDITIONAL IMPLEMENTATION DETAILS

In this section, we discuss additional implementation details of PAPSP and MA-PAPSP.

**Evaluation Sets.** Performance is evaluated on the test splits of the following datasets.

- *Selective classification:* Oxford Pets, Oxford Flowers (Nilsback & Zisserman, 2008) and UCF-101 (Soomro et al., 2012).
- *Selective image-text matching.* SugarCrepe (Hsieh et al., 2024), WinoGround (Thrush et al., 2022), What'sUp (Kamath et al., 2023), VL-Checklist (Zhao et al., 2022) and Foil (Shekhar et al., 2017).
- *Selective captioning:* MS-COCO (Lin et al., 2014) and Flickr-30K (Plummer et al., 2015).

**Implementation.** All experiments implemented using PyTorch (Paszke et al., 2019) and run on a single NVIDIA RTX 3090 GPU, using the pre-trained VLMs trained by OpenCLIP (Ilharco et al., 2021). The source codes of MA-PAPSP and PAPSP will be released upon publication.

**Algorithm.** The algorithm of MA-PAPSP is summarized in Algorithm A.1.

---

**Algorithm A.1** MA-PAPSP

1: **Input:** image-caption pair $\mathbf{z} = \{\mathbf{v}, \mathbf{w}\}$, retrieval set $\mathcal{R} = \{\mathbf{x}_i, \mathbf{y}_i\}_{i=1}^{N}$
2: **Output:** score $s$
3: Compute $\mathcal{N}(\mathbf{v})$ using equation 5.
4: Compute proxy embedding $\tilde{\mathbf{w}}(\mathbf{v})$ using equation 6
5: **if** negative captions are present **then**
6:     Compute $s$ using equation 7
7: **else**
8:     Generate $\mathcal{E}$ for $f(\mathbf{x})$ using LLM
9:     Compute $s$ using equation 7
10: **end if**
11: return $s$

---

## B COMPARISON WITH BASELINES

In this section, we compare MA-PAPSP to several baselines, including implementations of PAPSP with different SP-VLMs (OpenCLIP (Radford et al., 2021), EVA02 (Fang et al., 2024), SigLIP (Zhai et al., 2023)), other image-text alignment modules (GenEval (Ghosh et al., 2023), Aesthetics score (Wang et al., 2019)), retrieval-augmented CLIP (REACT (Liu et al., 2023b)), and finetuned

| Model | MA | Classification | | | Captioning | | | | ITM | | | | |
|---|---|---|---|---|---|---|---|---|---|---|---|---|---|
| | | | | | MS-COCO | | Flickr-30K | | | | | | |
| | | Flowers | Pets | UCF101 | Cider-N | Meteor | Cider-N | Meteor | SugarCrepe | WinoGround | What'sUp | VL-Check | Foil |
| **Small Models** | | | | | | | | | | | | | |
| OpenCLIP_B/16 | | 0.160 | 0.231 | 0.213 | 0.153 | 0.320 | 0.241 | 0.316 | 0.173 | 0.322 | 0.292 | 0.266 | 0.257 |
| | ✓ | **0.111** | **0.137** | **0.133** | **0.130** | **0.316** | **0.238** | **0.313** | **0.123** | **0.248** | **0.267** | **0.249** | **0.242** |
| EVA02_B/16 | | 0.109 | 0.220 | 0.206 | 0.139 | 0.318 | 0.240 | 0.317 | 0.171 | 0.264 | 0.272 | 0.262 | 0.251 |
| | ✓ | **0.109** | **0.135** | **0.127** | **0.132** | **0.318** | **0.235** | **0.312** | **0.108** | **0.236** | **0.264** | **0.233** | **0.238** |
| SigLIP_B/16 | | 0.093 | 0.211 | 0.154 | 0.142 | 0.317 | 0.237 | 0.311 | 0.162 | 0.258 | 0.262 | 0.258 | 0.245 |
| | ✓ | **0.077** | **0.171** | **0.116** | **0.121** | **0.312** | **0.235** | **0.310** | **0.079** | **0.228** | **0.214** | **0.210** | **0.213** |
| **Large Models** | | | | | | | | | | | | | |
| OpenCLIP_L/14 | | 0.119 | 0.182 | 0.151 | 0.146 | 0.318 | 0.240 | 0.312 | 0.149 | 0.258 | 0.266 | 0.254 | 0.246 |
| | ✓ | **0.109** | **0.128** | **0.141** | **0.138** | **0.297** | **0.235** | **0.298** | **0.127** | **0.208** | **0.226** | **0.217** | **0.198** |
| EVA02_L/14 | | **0.102** | 0.175 | 0.126 | **0.134** | 0.313 | 0.237 | 0.313 | 0.128 | 0.242 | 0.248 | 0.232 | 0.229 |
| | ✓ | 0.106 | **0.117** | **0.118** | 0.121 | **0.284** | **0.228** | **0.295** | 0.128 | **0.216** | **0.216** | **0.196** | **0.199** |
| SigLIP_SO-400M | | 0.074 | 0.169 | 0.113 | 0.136 | 0.311 | 0.229 | 0.307 | 0.078 | 0.208 | 0.202 | 0.204 | 0.197 |
| | ✓ | **0.063** | **0.114** | **0.088** | **0.109** | **0.286** | **0.219** | **0.297** | **0.062** | **0.196** | **0.192** | **0.192** | **0.189** |

Table A.1: AURC (↓) performance of PAPSP with and without memory augmentation (MA) for different visual language representation models.

SP-VLMs (SigLIP_B/16, SigLIP_SO-400M). Beyond this, we also consider models designed for selective prediction (LYP (Dancette et al., 2023), ReCoVERR (Srinivasan et al., 2024)), and LVLM-as-a-judge (Qwen-2.5-VL). Additionally, we analyze the inference latency of all methods. All experiments were conducted using the setup described in the main paper.

**PAPSP with SP-VLMs.** Table A.1 compares MA-PAPSP to PAPSP for different SP-VLMs, for all three tasks described in the main paper. The table supports the following conclusions. First, MA-PAPSP outperforms PAPSP consistently across tasks and datasets for both small and large SP-VLMs. Second, the gains of MA-PAPSP are larger for the larger SP-VLMs. This suggests that there is no saturation, *i.e.*even the largest VLRMs can benefit from the increase in representation stability and calibration afforded by MA-PAPSP.

| Methods | Classification | | | Captioning | | | | ITM | | | | |
|---|---|---|---|---|---|---|---|---|---|---|---|---|
| | | | | MS-COCO | | Flickr-30K | | | | | | |
| | Flowers | Pets | UCF101 | Cider-N | Meteor | Cider-N | Meteor | SugarCrepe | WinoGround | What'sUp | VL-Checklist | Foil |
| LYP | 0.216 | 0.221 | 0.226 | 0.152 | 0.342 | 0.236 | 0.323 | 0.158 | 0.229 | 0.228 | 0.232 | 0.225 |
| ReCoVERR | 0.225 | 0.216 | 0.223 | 0.142 | 0.336 | 0.253 | 0.327 | 0.167 | 0.252 | 0.249 | 0.247 | 0.242 |
| PAPSP-S | 0.093 | 0.211 | 0.154 | 0.142 | 0.317 | 0.237 | 0.311 | 0.162 | 0.258 | 0.262 | 0.258 | 0.245 |
| MA-PAPSP-S | 0.077 | 0.171 | 0.116 | 0.121 | 0.312 | 0.235 | 0.310 | 0.079 | 0.228 | 0.214 | 0.210 | 0.213 |
| PAPSP-L | 0.074 | 0.169 | 0.113 | 0.136 | 0.311 | 0.229 | 0.307 | 0.078 | 0.208 | 0.202 | 0.204 | 0.197 |
| MA-PAPSP-L | **0.063** | **0.114** | **0.088** | **0.109** | **0.286** | **0.219** | **0.297** | **0.062** | **0.196** | **0.192** | **0.192** | **0.189** |

Table B.1: AURC (↓) performance of PAPSP and MA-PAPSP to selective predictors.

**Selective Predictors.** Table B.1 compares MA-PAPSP to methods designed specifically for selective prediction (SP): LYP (Dancette et al., 2023) and ReCoVERR (Srinivasan et al., 2024). Phi-3.5-Instruct (3B) is used as the ReCoVERR LLM, and BLIP-2 is used as P-VLM for all methods. Shown in boldface are the overall best results, while the underlined results indicate where SP methods outperform PAPSP-S. PAPSP-L, and MA-PAPSP with both small and large model outperform both SP methods on 12/12 datasets. The gains are largest for MA-PAPSP-L. For instance, MA-PAPSP with small model has an AURC gain of 21 points and 18 Cider-N points over ReCoVERR on MS-COCO and Flickr-30K, respectively, while for MA-PAPSP with the largest model the gains become of 33 and 34 points, respectively. Note that this happens despite the fact that the SP models are computationally more intensive than MA-PAPSP and PAPSP.

**Image-Text Alignment Modules.** Table B.2 compares the performance of PAPSP and MA-PAPSP against state-of-the-art image-text alignment (ITA) models, including GenEval Score (Ghosh et al., 2023), Aesthetics Score (Wang et al., 2019), VQAScore (Lin et al., 2024)

| Methods | Classification | | | Captioning | | | | ITM | | | | |
|---|---|---|---|---|---|---|---|---|---|---|---|---|
| | | | | MS-COCO | | Flickr-30K | | | | | | |
| | Flowers | Pets | UCF101 | Cider-N | Meteor | Cider-N | Meteor | SugarCrepe | WinoGround | What'sUp | VL-Checklist | Foil |
| GenEval | 0.216 | 0.213 | 0.226 | 0.155 | 0.327 | 0.249 | 0.317 | 0.149 | 0.246 | 0.240 | 0.234 | 0.217 |
| Aesthetics | 0.212 | 0.211 | 0.225 | 0.155 | 0.322 | 0.242 | 0.318 | 0.153 | 0.241 | 0.241 | 0.227 | 0.221 |
| VQAScore | 0.211 | 0.207 | 0.217 | 0.146 | 0.321 | 0.241 | 0.312 | 0.146 | 0.240 | 0.236 | 0.226 | 0.217 |
| SeeTRUE | 0.214 | 0.213 | 0.171 | 0.158 | 0.316 | 0.251 | 0.411 | 0.153 | 0.242 | 0.236 | 0.234 | 0.223 |
| PaPSP-S | 0.093 | 0.211 | 0.154 | 0.142 | 0.317 | 0.237 | 0.311 | 0.162 | 0.258 | 0.262 | 0.258 | 0.245 |
| MA-PaPSP-S | 0.077 | 0.171 | 0.116 | 0.121 | 0.312 | 0.235 | 0.310 | 0.079 | 0.228 | 0.214 | 0.210 | 0.213 |
| PaPSP-L | 0.074 | 0.169 | 0.113 | 0.136 | 0.311 | 0.229 | 0.307 | 0.078 | 0.208 | 0.202 | 0.204 | 0.197 |
| MA-PaPSP-L | **0.063** | **0.114** | **0.088** | **0.109** | **0.286** | **0.219** | **0.297** | **0.062** | **0.196** | **0.192** | **0.192** | **0.189** |

Table B.2: AURC ($\downarrow$) for all three tasks. The table shows the comparison between PaPSP, MA-PaPSP and Image-text alignment modules.

and SeeTRUE (Yarom et al., 2023). Boldface highlights the best-performing models, while the underlined results indicate where ITA methods outperform PaPSP-S. PaPSP-L and MA-PaPSP with both small and large model outperform both ITA methods on 12/12 datasets. The gains of the larger versions of MA-PaPSP over ITA baselines are also more pronounced than for smaller models. Specifically, MA-PaPSP with the largest model outperforms the GenEval score by 44 points on MS-COCO and 30 points on Flickr-30K, under the Cider-N setting. This gain holds despite the fact that the ITA methods are significantly more computationally expensive than PaPSP and MA-PaPSP, as they rely on either visual-question answering with large vision-language models (LVLMs) or object detection combined with large language models (LLMs) to identify the presence of objects mentioned in the prompt. This again highlights the effectiveness of MA-PaPSP.

| LVLM-as-Judge | Captioning | | | | Classification | | | ITM | | | | |
|---|---|---|---|---|---|---|---|---|---|---|---|---|
| | MS-COCO | | Flickr-30K | | | | | | | | | |
| | Cider-N | Meteor | Cider-N | Meteor | Flowers | Pets | UCF-101 | SugarCrepe | Winoground | What'sUp | VL-checklist | Foil |
| | | | | | *LVLM-PaPSP* | | | | | | | |
| Qwen-VL-7B | 0.120 | 0.301 | 0.233 | 0.312 | 0.075 | 0.126 | 0.099 | 0.073 | 0.209 | 0.205 | 0.205 | 0.202 |
| Qwen-VL-72B | 0.115 | 0.294 | 0.227 | 0.305 | 0.070 | 0.120 | 0.094 | 0.068 | 0.203 | 0.199 | 0.199 | 0.196 |
| | | | | | *MA-PaPSP* | | | | | | | |
| SigLIP$_{SO-400M}$ | **0.109** | **0.286** | **0.219** | **0.297** | **0.063** | **0.114** | **0.088** | **0.062** | 0.196 | 0.192 | **0.192** | **0.189** |

Table B.3: Performance of MA-PaPSP as compared to LVLM-as-a-judge baselines.

**LVLM-as-a-judge.** We implemented a version of LVLM-as-judge, referred to as LVLM-PaPSP, which is comparable to PaPSP and MA-PaPSP. In this setup, an LVLM is tasked with generating an alignment score between the query image and its corresponding caption. This score is subsequently thresholded to facilitate selective prediction. Table B.3 presents the results for both the smaller (7B parameters) and larger (72B parameters) LVLMs from the Qwen-2.5-VL family. MA-PaPSP outperforms LVLM-PaPSP across all datasets and tasks, for both the 7B and 72B parameter models. These findings confirm the hypotheiss that LVLMs are not well-calibrated to produce reliable confidence scores for selective prediction.

| Model | Captioning | | | | Classification | | | ITM | | | | |
|---|---|---|---|---|---|---|---|---|---|---|---|---|
| | MS-COCO | | Flickr-30K | | | | | | | | | |
| | Cider-N | Meteor | Cider-N | Meteor | Flowers | Pets | UCF-101 | SugarCrepe | Winoground | What'sUp | VL-checklist | Foil |
| PaPSP | 0.136 | 0.311 | 0.229 | 0.307 | 0.074 | 0.169 | 0.113 | 0.078 | 0.208 | 0.202 | 0.204 | 0.197 |
| PaPSP-FT | 0.115 | 0.278 | 0.224 | 0.301 | 0.067 | 0.074 | **0.071** | 0.074 | 0.192 | 0.188 | 0.202 | **0.159** |
| MA-PaPSP | 0.126 | 0.288 | 0.229 | 0.307 | 0.062 | 0.076 | 0.078 | 0.066 | 0.192 | 0.189 | 0.207 | 0.167 |
| MA-PaPSP-FT | **0.104** | **0.234** | **0.217** | **0.288** | **0.060** | **0.068** | 0.072 | **0.064** | **0.188** | **0.172** | **0.201** | 0.162 |

Table B.4: Performance of MA-PaPSP as compared to PaPSP and PaPSP with finetuned SigLIP$_{SO-400M}$. MA-PaPSP is implemented with SigLIP$_{SO-400M}$.

**SP-VLM Finetuning.** We evaluated the benefits of fine-tuning the SP-VLM on the in-domain dataset for both PaPSP and MA-PaPSP. In all experiments, the SP-VLM used was SigLIP$_{SO-400M}$,

and the retrieval set for MA-PAPSP consisted of the in-domain dataset. The results are summarized in Table B.4, were boldface denotes the best results and underline indicates when PAPSP-FT outperforms MA-PAPSP without finetuning. A few conclusions are possible. First, while PAPSP-FT outperforms MA-PAPSP on 9/12 datasets, the differences are small: around 10 points on MS-COCO, around 5 points on Flickr, between 2 and 7 points for the classification tasks, and between 1 and 8 points for ITM. Note that PAPSP-FT, requires training the SP-VLM for each retrieval set, which is cumbersome and computationally intensive. For example, finetuning the SigLIP$_{\text{SO-400M}}$ VLRM on the MS-COCO train split using a single RTX-A4000 requires 20 hours of training. The fact that MA-PAPSP is competitive *without any training* is a sign of its effectiveness. Second, if the training data available for finetuning is small, the finetuned VLRM can overfit leading to weak AURC. MA-PAPSP is more robust to small data sizes. For instance, in the ITM tasks, which tend to have small datasets (e.g. 1K samples for SugarCrepe), MA-PAPSP outperforms or is almost identical to PAPSP-FT on 3/5 datasets. Third, the best overall results are obtained with PAPSP-FT, which has the top performance in 10/12 datasets, and a loss of at most 3 points in the remaining 2/12. This shows that, even if a fine-tuned model is available (PAPSP-FT), MA-PAPSP is still beneficial. In fact the gains of MA-PAPSP-FT over PAPSP-FT can be larger than those of MA-PAPSP over PAPSP. For example, on Flickr-30K, MA-PAPSP has no gain over PAPSP but MA-PAPSP-FT has gains between 7 and 13 points over PAPSP-FT. In summary, the decision to finetune the SP-VLM must take into account whether the gains of PAPSP-FT over MA-PAPSP justify the additional training complexity. However, once the finetuned VLRM is available there is almost always a benefit in augmenting it with MA-PAPSP.

| Methods | Time (ms) ↓ |
|---|---|
| PAPSP-S/L | 14.07/14.11 |
| Close-set Task (Classification, ITM) | |
| MA-PAPSP-L (P) | 14.87 |
| Open-set Task (Captioning) | |
| MA-PAPSP-L (w RB) (P+C) | 17.54 |
| MA-PAPSP-L (w SLM) (P+C) | 57.93 |
| MA-PAPSP-L + PP (w SLM) (P+C) | 25.85 |

Table B.5: Average inference times. P and C denote the two components of MA-PAPSP with RB or SLM, Proxy Embedding and Contrastive Score, respectively. PP denotes Pipeline parallelism over 2 GPUs.

**Per-sample Inference Latency.** Table B.5 presents a comparison of the inference time per sample across various methods. The top third of the table shows the inference times for PAPSP using both the small and large SP-VLMs, revealing that model size has minimal impact on latency. The middle third of the table displays the MA-PAPSP inference times for tasks, such as classification and ITM, that only require the computation of the proxy embedding (P). For these tasks, MA-PAPSP is comparable to PAPSP. The bottom third of the table shows the times for tasks like captioning, which involve both the proxy embedding and the contrastive score (P+C). This section highlights that the most time-consuming operation is the computation of contrastive scores, primarily due to the use of the LLM for generating hard negatives.

For captioning, MA-PAPSP is approximately three times slower than PAPSP. However, we note that several optimizations could significantly improve its speed. For instance, by employing pipeline parallelism across two RTX-3090 GPUs to parallelize the LLM and MA-PAPSP computations (denoted as MA-PAPSP-L+PP in the table), the overall inference time for MA-PAPSP can be reduced by more than half. Additionally, the nearest neighbor operation involved in proxy computation could be substantially accelerated using techniques such as Faiss (Douze et al., 2024), ScaNN (Guo et al., 2020), or indexing structures from classical retrieval methods (Lewis et al., 2020; Iscen et al., 2023a; Xie et al., 2023). The contrastive score computations could also be sped up significantly by replacing the LLM with rule-based sentence manipulations, as seen in prior work (Wang et al., 2024; Shekhar et al., 2017). While this approach could lead to slight degradation in AURC performance, it remains an avenue for future investigation. It is important to note that the computation of the retrieval set embeddings is a one-time operation and is not included in the times reported above, consistent

with standard retrieval methods. For reference, the per-sample embedding time (on RTX-3090) and storage for $CLIP_{B/16}$ are 10.361 ms and 2.274 KB, respectively.

**Per-sample Retrieval Latency with ScaNN.** We now present a comparison using ScaNN (Refahi et al., 2025), which is a recent deep-learning based approach to retrieval, and pipeline parallelism across 4 RTX-A6000 GPUs, which is a commonly used procedure in practical applications of retrieval. Detailed results for this setting are summarized in Table B.6. All the MA-PAPSP methods implement both the proxy embedding and contrastive score computation. The latter is implemented with either the rule-based (RB) approach or the small language model (SLM) as mentioned earlier.

| Methods | Time (ms) ↓ | MS-COCO | Flickr |
|---|---|---|---|
| MA-PAPSP with k-NN retriever | | | |
| MA-PAPSP-L (RB) | 49.54 | 0.109 | 0.219 |
| MA-PAPSP-L (SLM) | 58.85 | 0.103 | 0.216 |
| MA-PAPSP with ScaNN retriever | | | |
| MA-PAPSP-L (RB) | 37.54 | 0.109 | 0.219 |
| MA-PAPSP-L (SLM) | 45.85 | 0.103 | 0.214 |
| PAPSP with LVLM-as-judge | | | |
| Qwen-2.5-VL (72B) | 1563.82 | 0.115 | 0.227 |
| Qwen-2.5-VL (7B) | 49.38 | 0.120 | 0.233 |
| Qwen-2.5-VL (3B) | 27.64 | 0.147 | 0.259 |

Table B.6: Average inference times and performance of different MA-PAPSP models on captioning with out-of-domain retrieval set.

These experiments, support the following conclusions. First, while the selective prediction (SP) performance of MA-PAPSP is not affected by the use of different retrieval methods, the computation can be substantially improved. Using ScaNN reduces latency by more than 12ms, i.e. to about 75% of the kNN time. Second, for context, we compare the per-sample inference latency of MA-PAPSP and several LVLMs (LVLM-as-judge). MA-PAPSP has latency comparable to that of 7B parameter LVLM, but much better performance. In fact, it has substantially better performance than the 72B parameter LVLM, which is $32\times$ slower. We believe that this shows that MA-PAPSP is a very computationally efficient approach to selective prediction.

**Is performance of MA-PAPSP limited by SP-VLM?** Performance of MA-PAPSP, just like any other multimodal LLM as judge, is limited by SP-VLM being employed. Table B.6 shows that MA-PAPSP with SigLIP (a 1B-parameter model) as the SP-VLM is much faster *and* achieves better performance than much larger LVLMs, such as Qwen-2.5-VL (72B parameters). This shows that the SP-VLM is not a bottleneck. This finding is justified by the fact that the vision encoders of most LVLMs are quite similar to the SP-VLMs that we use. So, LVLMs suffer from exactly the same issues that the PAPSP model faces. Memory augmentation with equation 6 and contrastive captions 7 improves the representation of the SP-VLM and help overcome these issues. This type of improvement is difficult to achieve for LVLMs, where the encoder is inside the model and not accessible. Improving the encoder requires retraining the whole model. In any case, as the visual encoders of LVLMs improve, those encoders can themselves be used as SP-VLMs in MA-PAPSP. So, it is more likely that SP-VLMs will advance faster than LVLMs. In fact, we leverage this, *e.g.* by using the more recent SigLIP instead of CLIP as SP-VLM. Many LVLMs are still implemented with CLIP, which is a weaker representation model but not easy to replace. Our results show that MA-PAPSP is quite effective even for the state-of-art SigLIP representation model. To the best of our knowledge, there is no method in the literature whose SP performance is superior to MA-PAPSP, either using an external judge or not.

**Different Metrics.** In the main paper, all evaluations are based on AURC scores. Recent work in selective prediction (Traub et al., 2024) has introduced alternative evaluation metrics. Table B.7 compares MA-PAPSP and PAPSP using the Area Under the Generalized Risk Coverage (AUGRC). While the absolute values differ, both metrics lead to the same conclusions.

**Out-of-distribution detection.** For classification tasks, selective prediction is usually important to distinguish out-of-distribution (OOD) samples from in-distribution (ID) samples. To evaluate

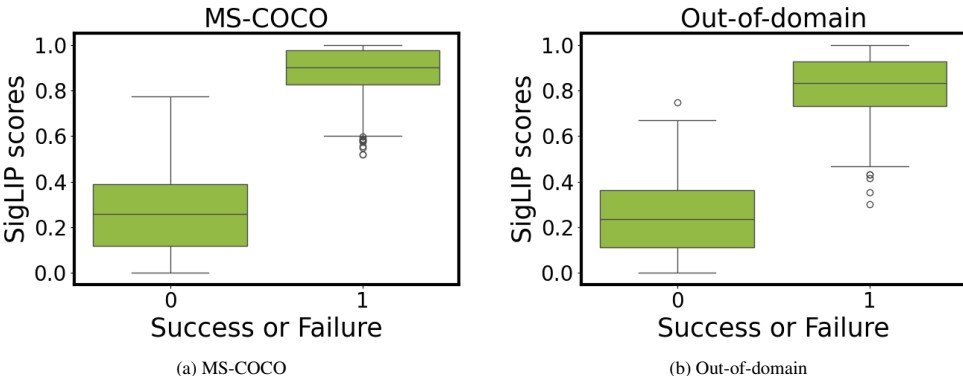

(a) MS-COCO                     (b) Out-of-domain

Figure B.1: Distribution of SigLIP scores when MA-PAPSP fails or succeeds on MS-COCO with the retrieval set noted at the top of the figure.

| Methods | Captioning | | | | Classification | | | ITM | | | | |
|---|---|---|---|---|---|---|---|---|---|---|---|---|
| | MS-COCO | | Flickr-30K | | Flowers | Pets | UCF101 | SugarCrepe | WinoGround | What'sUp | VL-Checklist | Foil |
| | Cider-N | Meteor | Cider-N | Meteor | | | | | | | | |
| AUGRC Metric (↓) | | | | | | | | | | | | |
| PAPSP-S | 0.207 | 0.331 | 0.224 | 0.365 | 0.106 | 0.185 | 0.124 | 0.189 | 0.293 | 0.311 | 0.271 | 0.272 |
| MA-PAPSP-S | **0.165** | **0.318** | **0.219** | **0.319** | **0.087** | **0.162** | **0.118** | **0.168** | **0.226** | **0.278** | **0.262** | **0.253** |
| PAPSP-L | 0.127 | 0.291 | 0.235 | 0.210 | 0.206 | 0.178 | 0.082 | 0.148 | 0.218 | 0.263 | 0.245 | 0.258 |
| MA-PAPSP-L | **0.119** | **0.266** | **0.221** | **0.184** | **0.166** | **0.148** | **0.058** | **0.102** | **0.196** | **0.198** | **0.202** | **0.194** |
| AURC Metric (↓) | | | | | | | | | | | | |
| PAPSP-S | 0.093 | 0.211 | 0.154 | 0.142 | 0.317 | 0.237 | 0.311 | 0.162 | 0.258 | 0.262 | 0.258 | 0.245 |
| MA-PAPSP-S | **0.077** | **0.171** | **0.116** | **0.121** | **0.312** | **0.235** | **0.310** | **0.079** | **0.228** | **0.214** | **0.210** | **0.213** |
| PAPSP-L | 0.074 | 0.169 | 0.113 | 0.136 | 0.311 | 0.229 | 0.307 | 0.078 | 0.208 | 0.202 | 0.204 | 0.197 |
| MA-PAPSP-L | **0.063** | **0.114** | **0.088** | **0.109** | **0.286** | **0.219** | **0.297** | **0.062** | **0.196** | **0.192** | **0.192** | **0.189** |

Table B.7: Selective prediction results on AUGRC (↓) for AURC (↓) metrics. S denotes the use of a small SP-VLM (SigLIP$_{B/16}$), while L denotes the larger SigLIP$_{SO-400M}$. In both cases, MA-PAPSP is implemented with the **out-of-domain** retrieval dataset.

.

MA-PAPSP in this setting, we considered the test splits of Oxford Pets, Oxford Flowers and UCF-101. In all cases, 80% of the classes were considered ID while the remaining 20% were considered OOD, and a classifer implemented for the ID classes. The largest class probability was used to implement selective prediction. Table B.8 summarizes the performance of PAPSP and MA-PAPSP for different SP-VLMs. For both small and large SP-VLMs, MA-PAPSP outperforms PAPSP in all datasets. In fact, MA-PAPSP with the small SP-VLM outperforms PAPSP with the large SP-VLM in all cases.

## C   ABLATIONS

This section contains more ablation results for MA-PAPSP.

| Methods | Flowers | Pets | UCF101 |
|---|---|---|---|
| PAPSP (SigLIP$_{B/16}$) | 0.100 | 0.112 | 0.157 |
| MA-PAPSP (SigLIP$_{B/16}$) | **0.067** | **0.108** | **0.112** |
| PAPSP (SigLIP$_{SO-400M}$) | 0.088 | 0.112 | 0.130 |
| PAPSP (SigLIP$_{SO-400M}$) | **0.054** | **0.101** | **0.119** |

Table B.8: AURC (↓) for OOD detection.

| | Captioning | | | | Classification | | | ITM | | | | |
|---|---|---|---|---|---|---|---|---|---|---|---|---|
| | MS-COCO | | Flickr-30K | | | | | | | | | |
| $K$ | Cider-N | Meteor | Cider-N | Meteor | Flowers | Pets | UCF-101 | SugarCrepe | Winoground | What'sUp | VL-checklist | Foil |
| 1 | 0.153 | 0.332 | 0.296 | 0.335 | 0.114 | 0.132 | 0.096 | 0.073 | 0.204 | 0.216 | 0.215 | 0.209 |
| 5 | 0.130 | 0.312 | 0.237 | 0.317 | **0.063** | **0.114** | **0.088** | 0.068 | 0.197 | 0.197 | 0.208 | 0.206 |
| 15 | **0.109** | **0.286** | **0.219** | **0.297** | 0.065 | **0.114** | 0.089 | **0.062** | **0.196** | **0.192** | **0.192** | **0.189** |
| 20 | 0.111 | 0.287 | 0.221 | 0.305 | 0.064 | **0.114** | 0.088 | 0.064 | **0.196** | **0.192** | 0.195 | 0.192 |

Table C.1: Neighborhood size $K$ for all tasks.

| | Captioning | | | | Classification | | | ITM | | | | |
|---|---|---|---|---|---|---|---|---|---|---|---|---|
| | MS-COCO | | Flickr-30K | | | | | | | | | |
| $\mathcal{R}$ Size/% | Cider-N | Meteor | Cider-N | Meteor | Flowers | Pets | UCF-101 | SugarCrepe | Winoground | What'sUp | VL-checklist | Foil |
| 20 | 0.138 | 0.314 | 0.231 | 0.321 | 0.063 | 0.114 | 0.088 | 0.074 | 0.209 | 0.265 | 0.256 | 0.248 |
| 40 | 0.122 | 0.310 | 0.226 | 0.323 | 0.063 | 0.114 | 0.088 | 0.074 | 0.196 | 0.192 | 0.192 | 0.189 |
| 60 | 0.116 | 0.292 | 0.221 | 0.312 | 0.063 | 0.114 | 0.088 | 0.072 | 0.204 | 0.192 | 0.192 | 0.189 |
| 80 | 0.111 | 0.289 | 0.220 | 0.312 | 0.063 | 0.114 | 0.088 | 0.064 | 0.196 | 0.192 | 0.194 | 0.192 |
| 100 | **0.109** | **0.286** | **0.219** | **0.297** | **0.063** | **0.114** | **0.088** | **0.062** | **0.196** | **0.192** | **0.192** | **0.189** |

Table C.2: Performance of MA-PAPSP (SigLIP$_{\text{SO-400M}}$) versus the percentage of the retrieval dataset $\mathcal{R}$ for all tasks. Retrieval set $\mathcal{R}$ is in-domain.

**Neighborhood size.** The performance of MA-PAPSP depends on the neighborhood size $K$ of (5), i.e. the size of set of texts retrieved from $\mathcal{R}$. Table C.1 shows an ablation for this parameter, for the captioning, image-text matching and classification tasks. Initially, as $K$ increases, the AURC decreases, indicating better performance for larger retrieval sets. Note, in particular the advantages of the soft nearest neighbor embedding implemented by proxy embeddings over a hard nearest neighbor embedding, where just the top nearest neighbor is retrieved from $\mathcal{R}$. This corresponds to using $K = 1$. For all tasks and datasets, the soft nearest neighbor operation has better performance. These results confirm the importance of proxy embeddings that fuse information from several image-text pairs relevant to the query. However, beyond a certain $K$, performance starts to degrade, with AURC rising. This is because larger retrieval sets start to include irrelevant or poor matches, and the retrieved proxy is less related to the query. Such behavior is fairly common for nearest neighbor approaches. For both tasks, the best AURC score is achieved in the neighborhood of $K = 15$, suggesting an optimal balance between retrieval size and performance. We use this value as the default for MA-PAPSP in all experiments.

**Size of Retrieval Set.** To analyze the effect of size of retrieval set on the performance of MA-PAPSP, we randomly sub-sampled the *in-domain retrieval set*. Table C.2 shows that, as dataset size decreases, the performance of MA-PAPSP can degrade for the captioning task. This is because the samples relevant (nearest neighbors) to the query are removed from the retrieval set. However, for classification and ITM, performance stays mostly unaltered for sizes as low as $40\%$. This shows that these tasks are much less sensitive to the dataset size.

**Retrieval variants.** Experiments were performed to compare the performance of the different retrieval variants of Table 1, on the captioning, image-text matching and classification task, using a neighborhood size $K = 15$ and contrastive scoring. Table C.3 shows that, for the above tasks, image to text retrieval (i2tr) achieves the best performance under all metrics. This is consistent with previous observations that the CLIP similarity score (cosine similarity in CLIP embedding space) is more reliable when its two arguments are text embeddings, suggesting a better conceptual organization of the CLIP embedding for text (Iscen et al., 2023b). Since, under i2tr, the retrieved proxy is a proxy caption, i2tr is expected to outperform i2ir, which returns an image proxy. On the other hand, the use of the predicted caption $f(\mathbf{x})$ as a query is expected not to perform well, since any captioning errors will induce the retrieval of image-caption pairs from $\mathcal{R}$ that are not relevant to the captioning of $\mathbf{x}$. This is true for other two tasks as well. It explains why the t2ir and t2tr variants do not perform well. Given these results, we use i2tr as the default retrieval implementation of MA-PAPSP. This variant is used in all other experiments of the paper.

**Effect of relevance threshold.** The evaluation of selective prediction requires the specification of threshold $\beta$ in (3) for the risk computation. We ablated the importance of this threshold by plotting

| | Captioning | | | | Classification | | | ITM | | | | |
|---|---|---|---|---|---|---|---|---|---|---|---|---|
| | MS-COCO | | Flickr-30K | | | | | | | | | |
| Variant | Cider-N | Meteor | Cider-N | Meteor | Flowers | Pets | UCF-101 | SugarCrepe | Winoground | What'sUp | VL-checklist | Foil |
| i2ir | 0.135 | 0.358 | 0.309 | 0.335 | 0.128 | 0.154 | 0.209 | 0.142 | 0.256 | 0.279 | 0.245 | 0.214 |
| t2ir | 0.152 | 0.344 | 0.264 | 0.342 | 0.116 | 0.118 | 0.164 | 0.116 | 0.242 | 0.266 | 0.242 | 0.208 |
| t2tr | 0.162 | 0.367 | 0.269 | 0.341 | 0.106 | 0.126 | 0.169 | 0.128 | 0.238 | 0.257 | 0.231 | 0.216 |
| **i2tr** | **0.109** | **0.286** | **0.219** | **0.297** | **0.063** | **0.114** | **0.088** | **0.062** | **0.196** | **0.192** | **0.192** | **0.189** |

Table C.3: Retrieval Variants for all tasks.

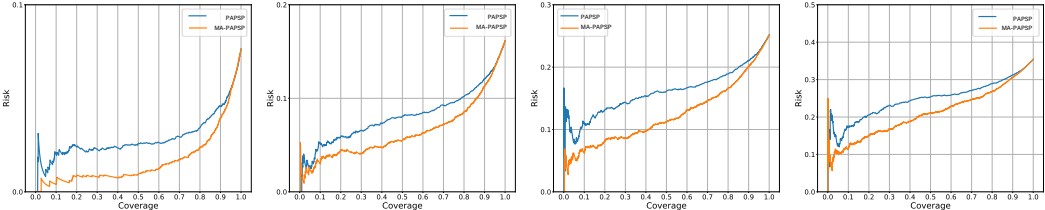

Figure C.1: Risk-Coverage curves of CLIP and PAPSP for selective image captioning under the Cider-N (N = 4) metric, for different values of the threshold $\beta$ used to define the risk. Left to right: $\beta = 0.2$, $\beta = 0.4$, $\beta = 0.6$, $\beta = 0.8$.

risk-coverage curves for different values of $\beta$. Figure C.1 shows an example for the Cider-N metric, for selective image captioning on the MS-COCO dataset, using MA-PAPSP and the OpenAI CLIP SP-VLM. While the four plots obtained with different values of $\beta$ are different, the relative behavior of PAPSP and MA-PAPSP curves is similar, with MA-PAPSP achieving substantially lower risk for most coverage values This shows that the exact value of $\beta$ is not critical. Note, however, that the same threshold should be used across experiments that involve comparison of AURC curves. We have followed this practice in all experiments.

| Variant | MS-COCO | | Flickr-30K | | Inference |
|---|---|---|---|---|---|
| | Cider-N | Meteor | Cider-N | Meteor | Time |
| MA-PAPSP without CC | | | | | |
| MA-PAPSP | 0.143 | 0.323 | 0.239 | 0.316 | - |
| MA-PAPSP w/ Rule-based CC | | | | | |
| Rule-Based | 0.113 | 0.301 | 0.222 | 0.301 | 4.72 ms |
| MA-PAPSP w/ LLM-based CC - Small LLM | | | | | |
| Qwen-2.5₃ᵦ | 0.117 | 0.306 | **0.213** | 0.304 | 42.26 ms |
| Phi-3.5₃ᵦ | 0.109 | 0.286 | 0.219 | 0.297 | 41.06 ms |
| MA-PAPSP w/ LLM-based CC - Large LLM | | | | | |
| Qwen-2.5₇ᵦ | 0.111 | 0.283 | **0.217** | 0.278 | 84.96 ms |
| Llama-3₈ᵦ | **0.108** | **0.277** | 0.219 | **0.276** | 92.65 ms |

Table C.4: Selective captioning AURC (↓) for different methods of contrastive component (CC).

**Contrastive component (CC).** For tasks like image-text matching and classification, negative samples are available by definition of the task. For image captioning, we compare two alternatives for generating contrastive examples: a *rule-based* approach and an *LM-based* approach. The rule-based method involves substituting objects, attributes, or verbs in the caption using a predefined set of rules. We used SpaCy to identify the part of speech—such as noun, adjective, or verb—and then replaced the identified word with another one from WordNet (Miller, 1995). In contrast, the LM-based method takes a given context (provided in E) and prompt and outputs a modified prompt with swapped objects, attributes, and verbs. Unlike the rule-based approach, the LM-generated captions are generally more coherent and grammatically correct as shown in E a. Table C.4 shows that, in addition to faster inference of the rule-based approach than that of LM-based methods, it is able to perform at par with even the implementation of MA-PAPSP with a LLM. For example MA-PAPSP with Phi-3.5 has just an improvement of 4 points over MA-PAPSP with rule based contrastive scores for MS-COCO with Cider-N setting. This indicates more effective generation

of contrastive samples. Further, while larger LLMs achieve better results, the differences are not very significant. For example, Llama-3 (8B parameter model) has an improvement of 1 point over Phi-3.5 (3B) for MS-COCO (Cider-N), but its inference time is almost double (92.65ms vs 43.06ms of Phi-3). Due to its better trade-off between performance and computation overhead, we use the rule-based approach in all our experiments.

## D  MORE RELATED WORKS

In this section, we talk about using more works that are related to MA-PAPSP.

**Foundation Models for Scoring.**  Visual language representation models, like CLIP (Radford et al., 2021), extract features from vision-language pairs and use contrastive learning (Ding et al., 2021) to learn a space where the two modalities are aligned. This enables the design of alignment scores based on simple operations, like cosine similarity, that underlie measures like the CLIP score (Hessel et al., 2021). Thresholding the CLIP score is a low-complexity solution for selective prediction, which we denote as PAPSP and use as baseline. MA-PAPSP improves on this baseline.

# E  QUALITATIVE RESULTS

We have presented the following two sets of qualitative figures - first, to show how the proxy component helps MA-PAPSP perform better than the baseline CLIP scores, we visualize some retrieved samples in Figures E.3, E.4, E.5 and E.6. Next, figure E.1 shows the prompt we were using to generate contrastive.

You are an intelligent and helpful AI assistant that can do the following task. Given an input context describing a scene, your task is to identify all the noun phrases in the sentence and then create a new sentence from all those noun phrases. The new sentence must meet the following four requirements:

1. The new sentence must be fluent and grammatically correct.
2. The new sentence must make logical sense.
3. The new sentence must not add any new noun phrases, but it must have all the noun phrases of the original sentence.

Here is one example:

Example 1: Sentence - A man and a woman are walking together.
New sentence: A man and a woman are talking to each other."
Explanation: There are two noun phrases in the sentence - man and woman. So, the new sentence contains those two noun phrases. It is different from the given context and does not include any extra noun phrases.

Generate 3 new sentences for this prompt. Do NOT give any explanation
Context - <prompt>
System:

Figure E.1: LLM Prompt used to generate negatives from a given prompt.

Examples of negative captions:

Main caption – A desk with a computer, a monitor, and a phone.
RB approach – A room with a man, a dog, and a airplane.
LM approach – A table with a glass of water and a fishbowl.

Main caption – A woman sitting on a bench on a city street.
RB approach – A boat sitting on a school on a rainbow.
LM approach – A man sitting on a chair in a quiet room.

Figure E.2: Example of negative captions generated by LM and RB.

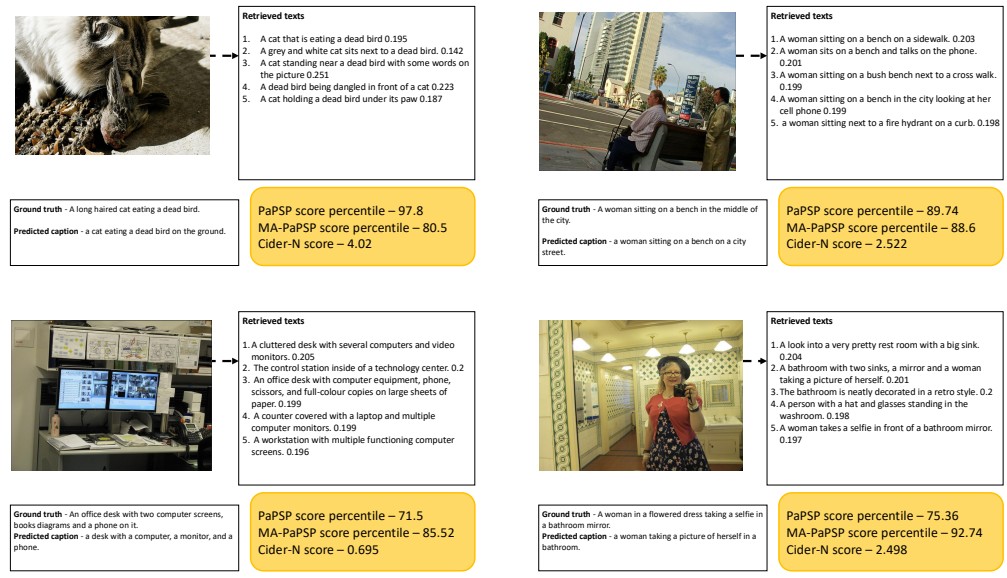

Figure E.3: This figure presents examples where both PAPSP and MA-PAPSP accept the inputs. Each example includes the following components: the image, its predicted caption, the ground-truth caption, the retrieved captions along with their normalized weights, PAPSP baseline scores, MA-PAPSP scores, and the ground-truth score (Cider-N).

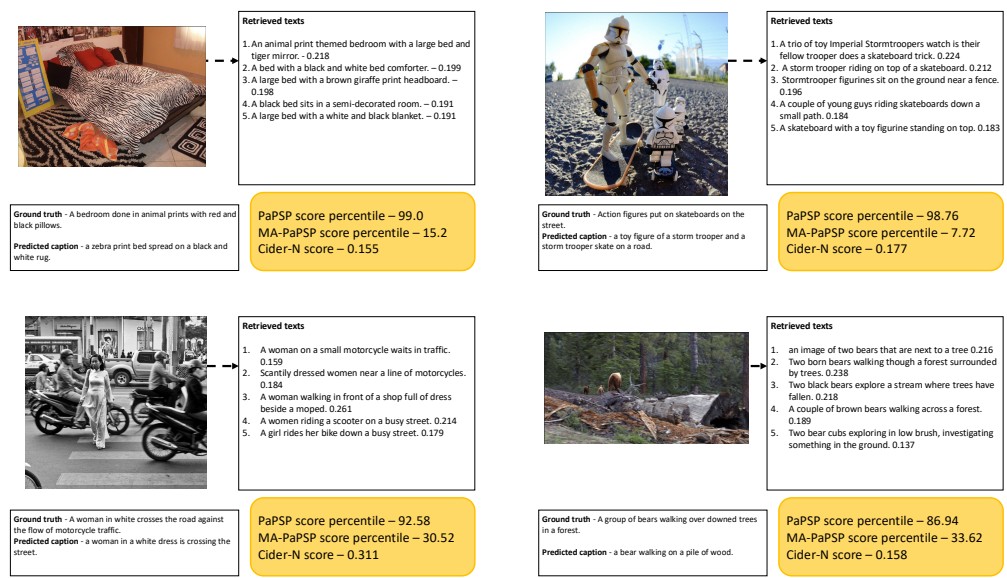

Figure E.4: This figure presents examples where PAPSP accepts and MA-PAPSP rejects the inputs. Each example includes the following components: the image, its predicted caption, the ground-truth caption, the retrieved captions along with their normalized weights, PAPSP baseline scores, MA-PAPSP scores, and the ground-truth score (Cider-N).

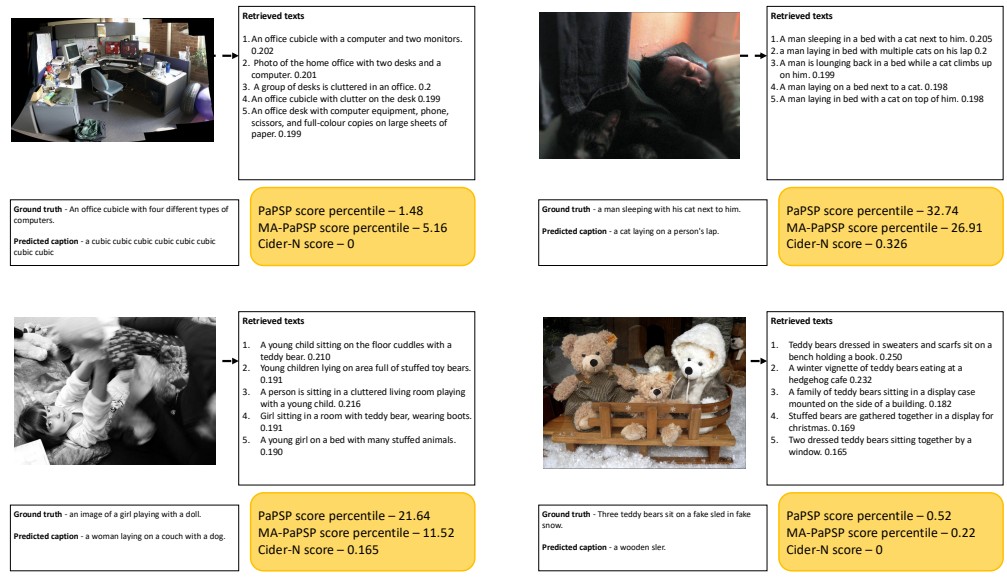

Figure E.5: This figure presents examples where both PAPSP and MA-PAPSP reject the inputs. Each example includes the following components: the image, its predicted caption, the ground-truth caption, the retrieved captions along with their normalized weights, PAPSP baseline scores, MA-PAPSP scores, and the ground-truth score (Cider-N).

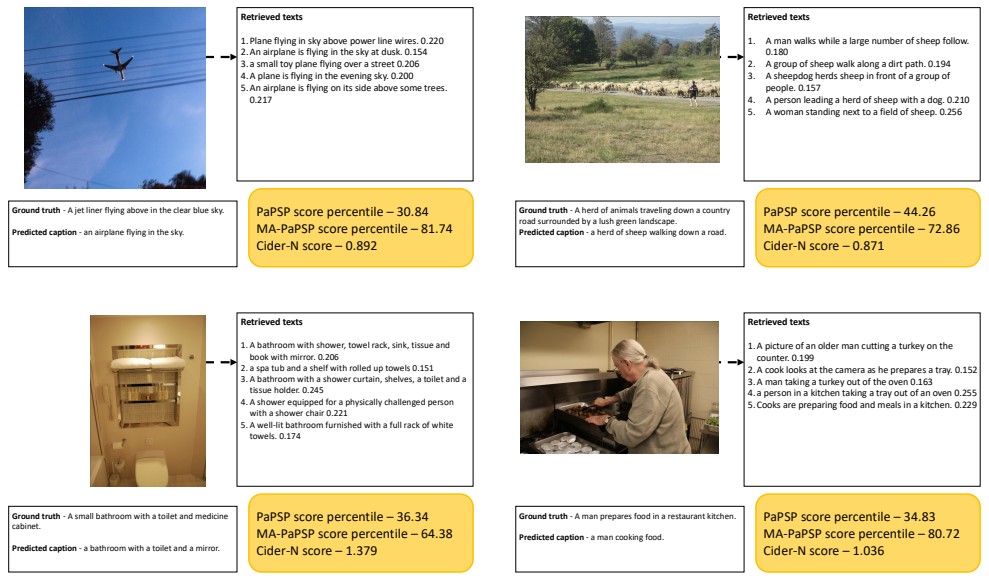

Figure E.6: This figure presents examples where PAPSP rejects and MA-PAPSP accepts the inputs. Each example includes the following components: the image, its predicted caption, the ground-truth caption, the retrieved captions along with their normalized weights, PAPSP baseline scores, MA-PAPSP scores, and the ground-truth score (Cider-N).