# OpenReview forum: "Leveraging Data to Say No: Memory Augmented Plug-and-Play Selective Prediction"
_ICLR.cc/2026/Conference — ICLR 2026 Poster_

### Official Review · Reviewer_8eZN · 2025-10-26

**Soundness:** 2
**Presentation:** 3
**Contribution:** 2
**Rating:** 4
**Confidence:** 3

**Summary:**

This paper proposes MA-PaPSP, a memory-augmented plug-and-play method for selective prediction in VLMs. It aims to improve upon standard PaPSP (e.g., CLIP-Score) by addressing its representation instability and poor score calibration. The method retrieves $k$-nearest neighbors from a dataset to create a more stable "proxy embedding" and uses a contrastive score for better calibration.

**Strengths:**

* The paper addresses a significant and practical problem: enabling training-free, plug-and-play selective prediction for foundation models, particularly for challenging open-set tasks like image captioning.

* The experimental evaluation is comprehensive. The method is validated across a clear taxonomy of VLM tasks (classification, ITM, captioning) and tested with various predictive VLMs (P-VLMs) of different scales.

**Weaknesses:**

* SP-VLM as Bottleneck: I am skeptical about the core premise. The paper focuses on augmenting a relatively weak SP-VLM (like CLIP/SigLIP) to judge a much stronger P-VLM (like Qwen-2.5-VL). As P-VLMs advance, the SP-VLM (the "judge") inherently becomes the bottleneck. The proposed MA mechanism seems like a complex workaround for this weak judge. A critical missing baseline is the "LLM-as-judge" approach: using a state-of-the-art LVLM (e.g., Qwen3VL or an API-based model) via prompt-engineering to directly score the output.
* Limited Novelty: The core technical contribution of the memory augmentation (MA) appears to be limited in novelty. The use of $k$-NN retrieval and weighted averaging (Eq. 6) to create a "proxy embedding" is a well-known ensemble and smoothing technique used to enhance robustness and accuracy. While its application to PaPSP is new, the underlying mechanism itself is not sufficiently novel.

**Questions:**

Conclusion: While the problem is well-motivated and the experiments are thorough, the two major weaknesses regarding the method's core premise (using a weak judge) and its technical novelty prevent a clear recommendation for acceptance.

I am willing to reconsider my score if the authors can convincingly address these two points, especially by providing a strong benchmark against a powerful "LLM-as-judge" baseline.

**Details Of Ethics Concerns:**

no concerns

---

> ### Author Response · Authors · 2025-11-20
> **Response to reviewer 8eZN (Part 1/3): SP-VLM is not a bottleneck even if P-VLMs advance.**
>
> We sincerely appreciate the reviewer’s thoughtful comments. It is gratifying to know that you acknowledge the significance of the problem our work addresses, consider our presentation of the research to be excellent, and view our experimental evaluation as comprehensive. I truly value your feedback!!
>
> We have responded to the questions you raised below and hope our explanations address all of your concerns. Should you have any further questions or require additional clarification, please feel free to reach out. We will make every effort to respond promptly.
>
> > **Q1. As P-VLMs advance, the SP-VLM (the ''judge") inherently becomes the bottleneck. The proposed MA mechanism seems like a complex workaround for this weak judge.**
>
> Our experiments show that this is actually not true, $\textit{i.e.}$ the use of an SP-VLM is $\textit{not}$ a bottleneck for $\textbf{\texttt{MA-PaPSP}}$. We specifically tested this, see for example, the results presented in the supplementary material L837–844. Essentially, we compared the performance of $\textbf{\texttt{MA-PaPSP}}$ to that of the LVLM-as-judge method, which is becoming increasingly popular for the evaluation of captioning results when complexity is not an issue. Table B.4 of the supplement shows that $\textbf{\texttt{MA-PaPSP}}$ has superior SP performance than LVLM-as-judge, across various LVLM sizes.
>
> We now extend this comparison in Table 1 of this rebuttal (see response 1/5 for reviewer 73qG), to also include latencies. The table shows that $\textbf{\texttt{MA-PaPSP}}$ with SigLIP (a 1B-parameter model) as the SP-VLM is much faster ${\it and}$ achieves better performance than much larger LVLMs, such as Qwen-2.5-VL (72B parameters). This shows that the SP-VLM is not a bottleneck. This finding is justified by the fact that the vision encoders of most LVLMs are quite similar to the SP-VLMs that we use. So, LVLMs suffer from exactly the same issues that the $\textbf{\texttt{PaPSP}}$ model faces. Memory augmentation with equation 6 and contrastive captions with equation 7 improve the representation of the SP-VLM and help overcome these issues.
>
> While it is true that P-VLMS will advance, a significant part of these advances are likely to be better visual encoders, which will also result in better SP-VLMs. Furthermore, it is much easier to improve the SP-VLM, which is external to the LVLM, than the LVLM encoder, an operation that requires retraining the entire model. So, it is more likely that SP-VLMs will advance faster than P-VLMs. In fact, we leverage this, $\textit{e.g.}$ by using the more recent SigLIP instead of CLIP as SP-VLM. Many LVLMs ($\textit{e.g.}$ BLIP, BLIP-2) are still implemented with CLIP, which is a weaker representation model but not easy to replace. Our results show that $\textbf{\texttt{MA-PaPSP}}$ is quite effective even for the state-of-art SigLIP representation model.
>
> > **Q2. The core technical contribution of the memory augmentation (MA) appears to be limited in novelty.**
>
> We disagree with the reviewer. While the individual components of $\textbf{\texttt{MA-PaPSP}}$ may not present significant novelty, the overall architecture, combining an external memory bank, proxy embeddings, and contrastive captions to address the calibration challenges of VLMs is, to our knowledge, both novel and innovative. To the best of our knowledge, no previous work applies this or a similar strategy to resolve calibration issues in LVLMs and produce confidence scores. In fact, we are not aware of any SP techniques for vision language problems like captioning. While SP is a long-studied problem, this has mostly been done in the domain of closed-set classification, using techniques that usually do not even generalize to open-set classification, let alone vision-language problems like captioning or image-text matching (ITM).
>
> Moreover, we believe that it is important to keep the $\textbf{\texttt{MA-PaPSP}}$ components simple, to minimize computational cost (see L890–911 in the supplementary section). Even though the components are simple by design, $\textbf{\texttt{MA-PaPSP}}$ outperforms **five different types of strong baselines**:
>
> 1. $\textbf{\texttt{PaPSP}}$ (see L387-403 and table A.1 in supplementary and tables 2,4,5 in main paper),
> 2. SOTA image-text alignment modules (see L803-825 and table B.1 in supplementary),
> 3. SOTA selective predictors (see L793-802 and B.2 in supplementary),
> 4. LVLM-as-judge (see L837-853 and table B.3 in supplementary) and
> 5. finetuned SP-VLM for $\textbf{\texttt{PaPSP}}$ (see L855-875 and table B.4 in supplementary).

---

> ### Author Response · Authors · 2025-11-20
> **Response to reviewer 8eZN (Part 2/3): Novelty in MA-PaPSP.**
>
> > **Q3. The use of -NN retrieval and weighted averaging (Eq. 6) to create a "proxy embedding" is a well-known ensemble and smoothing technique used to enhance robustness and accuracy. While its application to PaPSP is new, the underlying mechanism itself is not sufficiently novel.**
>
> Memory augmentation relies on an external dataset and a retrieval operation. This is a classic problem in computer vision, which has been the subject of hundreds of papers. Looking for novelty in this operation is extremely narrow minded.
>
> The main challenge for this paper is how to produce calibrated probabilities for captioning, a task which does not even involve a normalized score, e.g. a softmax type of operation. This needs to be achieved without sacrificing plug-and-play functionality, which rules out training of the external SP-VLM. A sophisticated retrieval operation, like the proxy embedding of eq 6 is important, but does not solve the problem, because it still does not produce probabilities. There is a need to generate a normalized score, which we achieve by generating negatives for the produced caption and scoring the caption with a contrastive score based on these negatives. Besides open-set tasks like captioning, this also makes the approach applicable to finite choice tasks, like classification or ITM. To the best of our knowledge, this $\textit{combination of methods}$ (retrieval and contrastive) has never been proposed for anything in computer vision, and we present the first paper to 1) implement selective prediction for open-set tasks like captioning and 2) go beyond the straightforward application of an external model, like CLIP or an LVLM, to implement the "as-judge" paradigm.
>
> For completeness, we discuss recent research works in memory-augmented multimodal learning, which do not even address the selective prediction problem for captioning.
>
> 1. [1,2,3] – These works focus on classification. They retrieve similar images using k-NN from an external dataset and apply voting, re-ranking, or dot-product scores to determine the final label. In contrast, $\textbf{\texttt{MA-PaPSP}}$ creates a proxy embedding from the retrieved samples (as described in equations 6 and 7) and computes a contrastive score with negative labels or captions it curates. Additionally, our method is generalized to both closed-set tasks (e.g., classification) and open-set tasks (e.g., captioning), whereas these methods are limited to closed-set classification tasks.
>
> 2. [4, 5] - These works focus on image captioning, where images samples are retrieved from a memory bank using k-NN and then passed through a language model to generate captions. They never address the problem of selective prediction or producing a calibrated probability score. Since the work of producing a caption is delegated to a language model, memory augmentation reduces to a simple nearest neighbors operation. Delegating the production of calibrated probabilities  to a language model is not effective, as our comparisons with the LVLM-as-judge clearly show. Furthermore,  the underlying approach is not even applicable to tasks like classification or ITM, where the output consists of discrete labels or options. $\textbf{\texttt{MA-PaPSP}}$ is able to handle this limitation because of its contrastive component, which beyond producing calibrated probabilities for captioning is also a natural fit for closed-set tasks like classification and ITM.
>
> 3. [6, 7] - These works retrieve similar samples from an external dataset using k-NN, pass them through CLIP’s image and text encoders, and then combine the resulting embeddings using either cross-attention or an MLP layer to obtain query embeddings.  This is a much more complex computation than the proxy embedding of $\textbf{\texttt{MA-PaPSP}}$. The whole network is then trained on a large dataset of natural images.  This is computationally expensive and ${\bf destroys}$ plug-and-play functionality, which is the main goal for $\textbf{\texttt{MA-PaPSP}}$. For example, these methods cannot be applied to specialized domains such as medical imaging without fine-tuning. It is not even clear that they can be applied to P-VLMs on which they were not trained. In contrast, $\textbf{\texttt{MA-PaPSP}}$ can be easily used with any P-VLM, as our results demonstrate, and applied to any domain by simply changing the retrieval dataset. These papers also do not consider the problem of selective prediction, which is the target application for $\textbf{\texttt{MA-PaPSP}}$. This and the fact that these methods are not open-sourced make direct comparisons with $\textbf{\texttt{MA-PaPSP}}$ impossible.

---

> ### Author Response · Authors · 2025-12-03
> **Response to reviewer 8eZN (Part 3/3): Novelty in MA-PaPSP.**
>
> **[Contd Q3]** Finally, we would comment that saying that "there is no novelty" in the paper because it relies on a combination of ``known techniques" is a comment that could be used to reject large swaths of the computer vision literature, e.g. almost all research in neural network architectures. After all, how was a residual connection new when the ResNet was introduced, or the use of attention or dot-products new when the transformer was first proposed? All architecture papers are based on new combinations of known techniques. And this applies to many other areas of computer vision. The reviewer is making a similar claim. Hence, we strongly disagree with the claim that there is no novelty in this paper.
>
> References -
>
> 1. Kengo Nakata et. al. Revisiting a knn-based image classification system with high-capacity storage. In ECCV, 2022.
> 2. Robert Geirhos et. al. Towards flexible perception with visual memory. arXiv preprint arXiv:2408.08172, 2024.
> 3. Thalles Silva et. al. Learning from memory: Non-parametric memory augmented self-supervised learning of visual features. 2024.
> 4. Zequn Zeng et. al. Meacap: Memory-augmented zero-shot image captioning. In CVPR, 2024.
> 5. Rita Ramos et. al. Smallcap: lightweight image captioning prompted with retrieval augmentation. In Proceedings of the IEEE/CVF Conference on Computer Vision and Pattern Recognition, pp. 2840–2849, 2023.
> 6. Ahmet Iscen et. al. Improving image recognition by retrieving from web-scale image-text data. In Proceedings of the IEEE/CVF Conference on Computer Vision and Pattern Recognition, pp. 19295–19304, 2023b.
> 7. Ahmet Iscen et. al. Retrieval-enhanced contrastive vision-text models. arXiv preprint arXiv:2306.07196, 2023a.

---

### Official Review · Reviewer_WFx9 · 2025-10-30

**Soundness:** 2
**Presentation:** 1
**Contribution:** 2
**Rating:** 4
**Confidence:** 3

**Summary:**

This paper presents a unified framework for addressing the selective prediction problem across various vision-language tasks, including classification, retrieval, and captioning. The goal is to derive a more reliable confidence score than the vanilla CLIP similarity score. To achieve this, the method first leverages neighborhood representations for arbitrary modalities to enhance representation stability, and then applies contrastive normalization to improve calibration.

**Strengths:**

The authors address the selective prediction problem, which aims to balance risk and coverage. This problem has been relatively underexplored in the context of vision-language models, particularly for open-set tasks such as captioning.

They propose a novel formulation that constructs neighborhood proxies using retrieval datasets, independent of the input or output modality. In addition, they present a unified framework that can be applied across different types of VLM tasks.

Through experiments, the authors further analyze the impact of retrieval datasets, which enhances the significance and credibility of their study.

**Weaknesses:**

The readability of this paper is unsatisfactory. Figure 1 is quite confusing; while it seems intended to illustrate the differences between classification and captioning in the proposed framework, the lines and arrows instead create misunderstanding. Figure 2 is difficult to interpret and contributes little to improving comprehension of the paper. Figure 4 is quite complex and lacks clear organization. The text in Figure 5 is too small, and the purpose of the figure itself is not clear enough.

In addition, the notations throughout the paper are generally hard to follow. Equation (6) is especially difficult to understand, possibly due to the ambiguous use of the symbols y and z. Overall, substantial revisions are needed before this work is ready for publication.

**Questions:**

Can the retrieval-based proxy proposed in this paper enhance open-set tasks in VLMs, such as captioning, beyond merely serving as a confidence score?

---

> ### Author Response · Authors · 2025-11-19
> **Response to reviewer WFx9 (Part 1/2): we will fix the equations and add intuitive explanations for the equations.**
>
> We are pleased to learn that you recognize the significance of the problem addressed in our work, appreciate the novelty of our approach, and find our analysis of the impact of retrieval datasets on our approach to be insightful.
>
> Below, we have addressed the queries you raised and hope that our responses fully clarify your concerns. If you have any more questions, please don't hesitate to let us know. We will try to promptly answer them as well.
>
> > **Q1. Can retrieval-based proxy proposed enhance open-set tasks in VLMs?**
>
> This is an interesting idea that we plan to pursue! Our current experiments show that proxy-based retrieval is effective for computing $\textit{external}$ confidence scores for P-VLMs. However, using these confidence scores to improve the P-VLM captioning performance requires further steps. For example, if the P-VLM is an LVLM, the scores could be used at test-time to elicit alternative responses from the latter, e.g. by using the external confidence score to guide the sampling of the generated sentence, rather than the internal LVLM score. Alternatively, the LVLM could be further trained, e.g. using reinforcement learning, to produce sentences of higher external confidence score. These are ideas that we plan to pursue in the future. However, they will require substantially more work and are beyond the scope of this paper.
>
> The main goal of this paper is to show that the external scores are accurate, something that can be objectively quantified by measuring selective prediction (SP) performance. The SP task is also important on its own because it has many important practical applications, e.g. avoiding hallucinations. Finally, the proposed $\textbf{\texttt{MA-PaPSP}}$ solution is applicable across a large diversity of P-VLMs, as shown in Table 4 of the main paper. Its use to improve the captioning performance itself is likely to be more P-VLM dependent, since different P-VLMs may use different mechanisms for sentence production.
>
> Overall, we believe that demonstrating success in the SP task should be enough for $\textbf{\texttt{MA-PaPSP}}$ to merit publication. In fact, other than $\textbf{\texttt{PaPSP}}$ or LVLM-as-judge, we are not aware of any approaches to implement SP for captioning in the literature. The results presented in the  paper and the results added in this rebuttal (please see replies to other reviewers) show that $\textbf{\texttt{MA-PaPSP}}$ is clearly better than these baselines, in terms of SP performance, latency, or both.

---

> ### Author Response · Authors · 2025-11-20
> **Response to reviewer WFx9 (Part 2/2): we will fix the equations and add intuitive explanations for the equations.**
>
> > **Q2. Notations throughout the paper are generally hard to follow.**
>
> The definitions of the symbols, such as $\textbf{x}$, $\textbf{y}$, and $\textbf{z}$, are provided in the main paper (L148–187 and L249–254). Explanations for equations (5–6) are included in the main paper (L201–215 and L249–269). The formulations of these equations draw from prior works on selective prediction [1,2,3,4,5]. Nonetheless, we offer a summary explanation of $\textbf{\texttt{MA-PaPSP}}$ below.
>
> - Equation (1) is the standard definition of selective prediction, It states that a selective model either outputs the model's predictions, $\textit{i.e.} f(\textbf{x})$, if confidence score $s(.)$ is higher than a given threshold or abstains from answering, $\textit{i.e.}$ outputs $\varnothing$. In $\textbf{\texttt{PaPSP}}$  and $\textbf{\texttt{MA-PaPSP}}$, this confidence score is computed by external SP-VLMs and is denoted as $s(.)$ in the paper.
>
> - Equation (5) is the neighborhood set curated by retrieving image-text pairs ($\textbf{x}^{R}$ and $\textbf{y}^{R}$ denote retrieved images and texts respectively) using a query $\textbf{q}$. Further, query $\textbf{q}$ can either be an image $\textbf{x}$ or text $\textbf{y}$. $\textbf{\texttt{MA-PaPSP}}$ uses input image $\textbf{x}$ for retrieval.
>
> - $\mathcal{I}_K$ in equation (5) is the set of indices of retrieved items.
>
> - Finally, equation (6) defines the proxy embeddings $\tilde{ {\bf \phi }}^e_\text{mod}({\bf q})$. The reviewer difficulty in understanding this equation probably stems  from missing the definitions of $\bf y$ in L251 and $\bf z$ in L251. Equation (6) defines the proxy embedding as a weighted average of embeddings of retrieved texts $\textbf{y}$, where the weights are determined by the similarity scores $\gamma(.,.)$ between the retrieved text ($\textbf{y}^{R}$) and query image ($\textbf{x}$), and normalized over all retrieved texts. We now realize that we did not mention that $\gamma$ is a cosine similarity. It can be implemented in any of the variants of Table 1, where "query" means $\bf q$ and "proxy" means ${\bf z}_i^R$ (again, defined in L251). This will be clarified in the final version of the paper.
>
> > **Q3. The readability of this paper is unsatisfactory.**
>
> Thank you so much for pointing this out! We are currently revising our figures to enhance clarity, including increasing the size of the figure captions for better readability. Additionally, we will refine the equations (Equations 1–6) in the main text and provide more intuitive explanations.
>
> > References
> 1. C. Chow. On optimum recognition error and reject tradeoff. IEEE Transactions on information theory (TIT),
> 16(1):41–46, 1970.
> 2. Yonatan Geifman and Ran El-Yaniv. Selective classification for deep neural networks. NeurIPS, 2017.
> 3. Pei Wang and Nuno Vasconcelos. Towards realistic predictors. In ECCV, 2018.
> 4. Emily Black, Klas Leino, and Matt Fredrikson. Selective ensembles for consistent predictions. In ICLR, 2022.
> 5. Tejas Srinivasan $\textit{et. al.}$. Selective” selective prediction”: Reducing unnecessary abstention in vision-language
> reasoning. arXiv preprint arXiv:2402.15610, 2024.

---

> > ### Comment · Reviewer_WFx9 · 2025-11-27
> >
> > Thanks for your clarification.

---

> ### Author Response · Authors · 2025-11-22
> **Thank you for reviewing our paper.**
>
> Dear Reviewer WFx9,
>
> Thank you for your review. We noticed that **all** of your comments focus on formatting issues in the paper, so we wanted to kindly ask whether you had any questions about the pipeline architecture or if you would like us to conduct additional experiments. We would be happy to clarify or expand on any technical aspects.
>
> As the paper is currently under ICLR review, we would be quite disappointed if the primary reason for a weak rejection were limited to formatting concerns. From the strengths and summary sections of your review, it appears that you understood the core contributions despite the presentation issues, and we appreciate that. However, we were surprised that the overall score seems to be heavily influenced by formatting alone, especially given that the other three reviewers evaluated the presentation positively.
>
> We respectfully request that you consider revisiting the paper with a focus on the technical contributions, and we would greatly appreciate any additional feedback you may have on the substance of our work.
>
> Best regards,
>
> Authors

---

> ### Author Response · Authors · 2025-11-25
> **Reminder for rebuttal.**
>
> Dear Reviewer WFx9,
>
> We appreciate your feedback during the first round of review. Since the discussion phase ends on Dec 3, we would like to know whether we have addressed all the issues, and we would greatly welcome any additional feedback or suggestions you may have. If all the concerns have been successfully addressed, please consider raising the scores after this discussion phase.
>
> Best regards,
>
> Authors

---

> > ### Comment · Reviewer_WFx9 · 2025-11-27
> >
> > Thanks for your response. With the clarification you provided on the presentation, I have carefully reviewed the paper again and now have a deeper understanding. I would like to raise several further concerns as follows.
> >
> > > Proxy Embeddings. To overcome the instability of the pre-trained embeddings, MA-PaPSP estimates the ground-truth embedding of a query q by a proxy embedding.
> >
> > Is there any possibility that the estimation error introduced by leveraging an external dataset could adversely affect calibration? In many cases, we can retrieve only *similar* data—and sometimes even that is challenging when the query domain is rare. It is not clear that a “ground-truth’’ embedding can always be reliably constructed from external data. I would like to see more analysis and discussion on this point.
> >
> > > MA-PaPSP can be seen as a new instantiation of memory-augmented approaches recently shown successful for various applications (Nakata et al., 2022; Zeng et al., 2024; Geirhos et al., 2024; Silva et al., 2024). Like these, it is a training-free method that instead leverages an external dataset to enhance SP-VLM functionality.
> >
> > The distinctions between MA-PaPSP and these prior works—beyond the fact that the augmentation technique here is applied to calibration—are not sufficiently discussed.
> >
> > My main concern is that this work essentially presents an application of external datasets to the selective prediction problem. It is not clear to me how this approach differentiates itself from the broader body of memory-augmented methods, especially since the performance gains appear to arise primarily from improved image recognition capability (i.e., a stronger CLIP score). Such improvements are already known to be achievable using external datasets with methodologies similar to those described in this paper, making the contribution feel less novel.
> >
> > For this reason, I asked whether you could provide deeper insights into how memory might be used to directly augment the captioning task itself, which I believe would be more compelling and informative. Additional discussion on how calibration, in particular, benefits from memory augmentation—especially in contrast to classification (i.e., the different mechanisms through which memory augmentation has impact)—would help clarify the significance of this work.

---

> ### Author Response · Authors · 2025-12-03
> **Response to reviewer WFx9 (Part 1/4)**
>
> > **Q1. Additional discussion on how calibration, in particular, benefits from memory augmentation—especially in contrast to classification (i.e., the different mechanisms through which memory augmentation has impact)—would help clarify the significance of this work.**
>
> $\textbf{\texttt{MA-PaPSP}}$ differs from RAG-based methods for classification in two main ways.
>
> First, classification is a much simpler problem, which makes retrieval augmentation much simpler to implement. Classification is a closed-set task, meaning that the number of labels is finite and relatively small and the labels are known, e.g. the ImageNet classes. Hence, memory augmentation can be performed by simple label voting. Just retrieve the closest images in the retrieval dataset, count their labels and pick the one with most votes. This is  the procedure used in [1,2]. $\textbf{\texttt{MA-PaPSP}}$ goes well beyond this. The goal is exactly to focus on open-set tasks, such as captioning and Image-Text Matching (ITM), where the "label" vocabulary is unbounded, e.g the space of natural language sentences, and there is not even a definition of what labels are, e.g. two different sentences can mean the same thing, and very similar sentences can have opposite meaning (e.g. by simple introduction of the word "not"). In fact, in ITM, the task is to choose between two similar sentences, picking the one that holds for the image. The methods used for closed-set classification do not even start to generalize to this context. For example, the retrieval component is much more trivial, there is no need consider cross-modal retrieval, such as the variants of Table 1, there is no need for sophisticated proxy embedding computations like equation 6 (simply retrieve the nearest image feature vectors), there is no need to produce negative hypothesis for contrastive scoring like equation 7-8 (since all the label set is fixed and all labels are known). Note that, because there is no predefined set of "other" labels to be used in a softmax type operation, open-set language retrieval does not even produce a normalized score, let alone calibrated probabilities. Before our work, it was not even clear that this would be feasible.  For example, it does not make sense to compare the performance of $\textbf{\texttt{PaPSP}}$ to those of the existing retrieval augmented classification methods because they simply $\textit{cannot support}$ tasks like captioning or ITM, let alone selective prediction for those tasks.
>
> Second, $\textbf{\texttt{MA-PaPSP}}$ aims to be training-free and plug-and-play, whereas previous methods such as [3,4,8] often leverage training to fuse retrieved samples and obtain the final score. This makes them specific to the models they are trained on, both the P-VLM and the SP-VLM, is computationally intensive, makes the SP-VLM difficult to apply to specialized domains, such as medical imaging, where fine-tuning may be needed to achieve optimal performance. Furthermore, these approaches have not been proposed for selective prediction and ignore some of the hardest issues of the latter, e.g. probability calibration. In contrast, our approach, $\textbf{\texttt{MA-PaPSP}}$, can directly leverage off-the-shelf CLIP models to generate improved SP scores, as demonstrated in lines 430-466 of the main paper.
>
> In summary, to address the first challenge, we introduced the improved retrieval (even cross-modal) and proxy computations of equation 6, a contrastive pipeline that generates a fixed set of negative captions based on the output caption and uses a softmax score to compute a confidence score. These negative captions are very similar to the output caption with very minor changes. For the second issue, we developed a fast-to-compute proxy embedding using the retrieved samples. Our experiments, as detailed in the main paper and supplementary, show that these two simple techniques significantly improve AURC scores and outperform $\textbf{five}$ different state-of-the-art baselines. Moreover, our approach is computationally efficient, retains its plug-and-play nature, is applicable to open-set tasks like captioning, and produces scores that are useful for selective prediction. No other method in the literature has most of these properties, let alone all four of them.

---

> ### Author Response · Authors · 2025-12-03
> **Response to reviewer WFx9 (Part 2/4)**
>
> > **Q2. Is there any possibility that the estimation error introduced by leveraging an external dataset could adversely affect calibration?**
>
> This is, of course always possible, in the same sense that any machine learning system may be faced with out-of-distribution data and fail. However, we have tested extensively this question and present results for various such settinsg, e.g. applying $\textbf{\texttt{MA-PaPSP}}$ to specialized domains. We provide a detailed discussion on this topic in L404-423 and L430-466 of main paper, and have acknowledged this limitation in L481-485 of section 5. Our results show that  this is a $\textbf{not}$ a $\textit{significant}$ challenge, or at least that it is not more of a significant challenge than that posed to competing approaches. For example, we compare the SP performance of $\textbf{\texttt{MA-PaPSP}}$ and LVLM-as-judge when the target domains are medical imaging datasets in Table 3 (see response 3/3 to tRii). These are be out-of-domain for both $\textbf{\texttt{MA-PaPSP}}$ (using SigLIP as SP-VLM and a generic retrieval set like CC12M) and most baselines.  A few conclusions can be  drawn.
>
> First, for $\textbf{\texttt{MA-PaPSP}}$, it is much more important to use a retrieval set from the target domain than a SP-VLM of the target domain. With the domain retrieval sets, SigLIP performs nearly as well as BioMedCLIP, which is trained on medical images.
>
> Second, for retrieval sets, "target domain" is a somewhat broad concept. As can be seen in the table, using as retrieval set CheXpert, MIMI-CXR, or both has similar performance, independently of the target dataset. This suggests that, as long as the retrieval and target datasets have some overlap, $\textbf{\texttt{MA-PaPSP}}$ performs well, which $\textit{reduces}$ the challenge of finding the right dataset.
>
> Third, even when combined with an out-of-domain retrieval set, the performance of $\textbf{\texttt{MA-PaPSP}}$ drops to the level of the performance of the LVLM-as-judge. Note that, in these experiments, the out-of-domain retrieval set is CC12M, which most practitioners would consider clearly out of domain, even $\textit{unrelated}$ to this task (medical imaging). So, this is a $\textit{worst case}$ result for $\textbf{\texttt{MA-PaPSP}}$. Even in this case, $\textbf{\texttt{MA-PaPSP}}$ is only slightly inferior to the LVLM-as-judge.
>
> In general, however, it is clear that no method, including the much more expensive LVLM-as-judge, performs well on clearly out-of-domain data. Hence, we disagree with the statement that having a memory distribution is just a challenge. On the contrary, the results of Table 3 (see response 3/3 of reviewer tRii) show that it also provides an $\textit{opportunity}$. Consider a practitioner faced with the setting of the table. In the LVLM-as-judge paradigm, the practitioner will have to find an $\textit{in-domain LVLM}$. This is likely to be much harder than finding an $\textit{in-domain dataset}$. For example, a medical clinic is likely to have data from similar exams but no capacity to train its own LVLM. Even if there is a medical LVLM that can be used as judge, that same LVLM  can be used to caption the data collected in the clinic to produce an in-domain retrieval set. As seen in Table 1 (see response 1/5 of reviewer 73qG), using $\textbf{\texttt{MA-PaPSP}}$ with this dataset is much more efficient than using the LVLM-as-judge solution. This shows that the use of a memory bank is also an $\textit{advantage}$ of $\textbf{\texttt{MA-PaPSP}}$, making it $\textit{a much more flexible and adaptable solution}$ than, for example, the LVLM-as-judge, without loss of performance  in comparable settings.
>
> > **Q3. It is not clear that a “ground-truth’’ embedding can always be reliably constructed from external data. I would like to see more analysis and discussion on this point.**
>
> We provided a detailed discussion on this topic in L404-423 and L430-466 of main paper. The main reason why a “ground-truth’’ embedding could not be reliably constructed from external data is a mismatch between the retrieval dataset and the target domain. As discussed above, we show that not even the SP-VLM has to be trained on the target domain. It suffices that the retrieval dataset is in the target domain. We believe that this is very strong empirical validation of the robustness of the computation of the "groundtruth" embedding. In any case, this is not a more significant challenge than that posed to competing approaches, as also discussed above.

---

> ### Author Response · Authors · 2025-12-03
> **Response to reviewer WFx9 (Part 3/4)**
>
> > **Q4. It is not clear to me how this approach differentiates itself from the broader body of memory-augmented methods.**
>
> The primary aim of our paper is to demonstrate how selective prediction performance can be enhanced by augmenting CLIP scores with external memory and contrastive scoring. This is the whole premise of this work. We don't intend to propose a brand new model for multimodal learning. We did not delve deeply into these works because they address different problems, and their solutions are tailored to specific tasks. These works differ from each other because of the way they fuse the retrieved samples to the main query. In that sense, equations 6-8 are novel and has not been used yet in any computer vision works to the best of our knowledge. In contrast, our approach is designed to improve selective prediction across a broader range of tasks.
>
> Below, we highlight how $\textbf{\texttt{MA-PaPSP}}$ differs from prior memory augmentation methods:
>
> 1. [1,2,3] – These works focus on classification. They retrieve similar images using k-NN from an external dataset and apply voting, re-ranking, or dot-product scores to determine the final label. In contrast, $\textbf{\texttt{MA-PaPSP}}$ creates a proxy embedding from the retrieved samples (as described in equations 6 and 7) and computes a contrastive score with negative labels or captions it curates. Additionally, our method is generalized to both closed-set tasks (e.g., classification) and open-set tasks (e.g., captioning), whereas these methods are limited to closed-set classification tasks.
>
> 2. [4, 5] - These works focus on image captioning, where images samples are retrieved from a memory bank using k-NN and then passed through a language model to generate captions. They never address the problem of selective prediction or producing a calibrated probability score. Since the work of producing a caption is delegated to a language model, memory augmentation reduces to a simple nearest neighbors operation. Delegating the production of calibrated probabilities  to a language model is not effective, as our comparisons with the LVLM-as-judge clearly show. Furthermore,  the underlying approach is not even applicable to tasks like classification or ITM, where the output consists of discrete labels or options. $\textbf{\texttt{MA-PaPSP}}$ is able to handle this limitation because of its contrastive component, which beyond producing calibrated probabilities for captioning is also a natural fit for closed-set tasks like classification and ITM.
>
> 3. [6, 7] - These works retrieve similar samples from an external dataset using k-NN, pass them through CLIP’s image and text encoders, and then combine the resulting embeddings using either cross-attention or an MLP layer to obtain query embeddings.  This is a much more complex computation than the proxy embedding of $\textbf{\texttt{MA-PaPSP}}$. The whole network is then trained on a large dataset of natural images.  This is computationally expensive and ${\bf destroys}$ plug-and-play functionality, which is the main goal for $\textbf{\texttt{MA-PaPSP}}$. For example, these methods cannot be applied to specialized domains such as medical imaging without fine-tuning. It is not even clear that they can be applied to P-VLMs on which they were not trained. In contrast, $\textbf{\texttt{MA-PaPSP}}$ can be easily used with any P-VLM, as our results demonstrate, and applied to any domain by simply changing the retrieval dataset. These papers also do not consider the problem of selective prediction, which is the target application for $\textbf{\texttt{MA-PaPSP}}$. This and the fact that these methods are not open-sourced make direct comparisons with $\textbf{\texttt{MA-PaPSP}}$ impossible.
>
> There are three key limitations associated with using either of the techniques mentioned above for fusing retrieved samples with the main query. First, they are not capable of addressing all the tasks outlined in our taxonomy. Second, they necessitate some level of training, which usually compromises the plug-and-play functionality at the core of $\textbf{\texttt{MA-PaPSP}}$ and can make implementation challenging in specialized domains without fine-tuning. Third, none of these pipelines address selective prediction or provide a calibrated confidence score, which is the main technical challenge for $\textbf{\texttt{MA-PaPSP}}$.

---

> ### Author Response · Authors · 2025-12-03
> **Response to reviewer WFx9 (Part 4/4)**
>
> > **Q5. The performance gains appear to arise primarily from improved image recognition capability (i.e., a stronger CLIP score).**
>
> We partly agree with the reviewer. It is true that the retrieval augmented operation implemented by $\textbf{\texttt{MA-PaPSP}}$ improves the performance of the SP-VLM. However, the gains are larger than just using a "stronger CLIP score". Our work demonstrates that further performance improvements can be achieved by leveraging relevant samples from an external dataset to compute a proxy embedding and applying a contrastive score. We discuss this in greater detail in Table 2 and lines 387–403 of the main paper, as well as in Table A.1 and lines 777–782 of the supplementary material.
>
> In fact, we make comparisons between models with ($\textbf{\texttt{MA-PaPSP}}$) and without ($\textbf{\texttt{PaPSP}}$) retrieval augmentation across SP-VLM sizes. We found that $\textbf{\texttt{MA-PaPSP}}$ with a smaller CLIP model can outperform or match the performance of $\textbf{\texttt{MA-PaPSP}}$ with a larger CLIP model, which has a stronger CLIP score. For example, as shown in Table A.1, $EVA02_{\text{B/16}}$ either outperforms or performs similarly to $\text{EVA02}_{\text{L/14}}$. The same conclusions hold for OpenCLIP and SigLIP models. Additionally, the ablation study in Table 6 of the main paper highlights the performance gains when these components are included. In summary, while retrieval augmentation improves the performance of the SP-VLM, the gains of $\textbf{\texttt{MA-PaPSP}}$ exceed straightforward approaches to improve the CLIP score.
>
> References -
> 1. Kengo Nakata et. al. Revisiting a knn-based image classification system with high-capacity storage. In ECCV, 2022.
> 2. Robert Geirhos et. al. Towards flexible perception with visual memory. arXiv preprint arXiv:2408.08172, 2024.
> 3. Thalles Silva et. al. Learning from memory: Non-parametric memory augmented self-supervised learning of visual features. 2024.
> 4. Zequn Zeng et. al. Meacap: Memory-augmented zero-shot image captioning. In CVPR, 2024.
> 5. Rita Ramos et. al. Smallcap: lightweight image captioning prompted with retrieval augmentation. In Proceedings of the IEEE/CVF Conference on Computer Vision and Pattern Recognition, pp. 2840–2849, 2023.
> 6. Ahmet Iscen et. al. Improving image recognition by retrieving from web-scale image-text data. In Proceedings of the IEEE/CVF Conference on Computer Vision and Pattern Recognition, pp. 19295–19304, 2023b.
> 7. Ahmet Iscen et. al. Retrieval-enhanced contrastive vision-text models. arXiv preprint arXiv:2306.07196, 2023a.
> 8. Alexander Long et. al. Retrieval augmented classification for long-tail visual recognition. In Proceedings of the IEEE/CVF conference on computer vision and pattern recognition, pp. 6959–6969, 2022.

---

### Official Review · Reviewer_tRii · 2025-10-31

**Soundness:** 3
**Presentation:** 3
**Contribution:** 3
**Rating:** 6
**Confidence:** 4

**Summary:**

This paper introduces MA-PaPSP, a lightweight and training-free method for selective prediction in vision-language models. It addresses the instability and poor calibration of standard similarity scores by an external memory and a calibration algorithm.

**Strengths:**

1. Problem Formulation and Framework: The formulation of the Plug-and-Play Selective Prediction (PaPSP) problem for a taxonomy of VLM tasks, especially open-set scenarios like captioning, is timely and meaningful.
2. Thorough Analysis: The paper provides a comprehensive analysis that successfully identifies and validates the core challenges of representation instability and score miscalibration in baseline methods.
3. Strong Generalization: The proposed MA-PaPSP method demonstrates impressive generalization capabilities across a wide range of tasks including specialized domains, provided a suitable memory bank is available.

**Weaknesses:**

1. The paper's claim of being a "lightweight" solution is somewhat challenged by its reliance on a massive external memory bank (e.g., 15M image-text pairs). The computational overhead and storage cost of performing nearest-neighbor retrieval from such a large database during inference should be more thoroughly discussed. This overhead could impact the method's practical deployment in latency-sensitive or resource-constrained environments, and a comparison of inference time against baselines like LLM-as-Judge would strengthen this analysis.
2. Dependence on Memory Quality and Practical Limitations: Model performance is highly dependent on the domain relevance of the memory bank, particularly in open-set settings. The necessity of procuring a memory distribution that aligns well with the target domain at test-time presents a significant practical challenge. This dependency could limit the method's applicability in scenarios where such curated data is unavailable.

**Questions:**

Was any analysis conducted to quantify the distributional similarity between the memory banks and the target evaluation datasets? Such an analysis would greatly strengthen the claims about why certain memory banks outperform others and provide guidance for memory selection in practice.

---

> ### Author Response · Authors · 2025-11-20
> **Response to reviewer tRii (Part 1/3): $\textbf{\texttt{MA-PaPSP}}$ is indeed a lightweight module.**
>
> We would like to sincerely thank the reviewer for their valuable comments. We are also pleased to learn that you recognize the significance of the problem addressed in our work, appreciate the novelty of our approach, and find our analysis of the impact of retrieval datasets on our approach to be insightful.
>
> Below, we have addressed the queries you raised and hope that our responses fully clarify your concerns. If you have any more questions, please don't hesitate to let us know. We will try to promptly answer them as well.
>
> > **Q1. The computational overhead and storage cost of performing nearest-neighbor retrieval from such a large database during inference should be more thoroughly discussed.**
>
> This is a non-trivial question because the computation overhead can be significantly reduced by using various techniques that are popular in the retrieval literature but not the focus of the paper. We provided some discussion in the paper supplement, $\textit{e.g.}$ Table B.5 and discussion starting in L890. However, all results reported to the implementation of retrieval with k-NN and the classic Ball Tree algorithm [1], which is not close to the state of the art in terms of computation. We now present a comparison using ScaNN [2], which is a recent deep-learning based approach to retrieval, and pipeline parallelism across 4 RTX-A6000 GPUs, which is a commonly used procedure in practical applications of retrieval. Detailed results for this setting are summarized in Table 1 (response 1/5 to reviewer 73qG). All the $\textbf{\texttt{MA-PaPSP}}$ methods implement both the proxy embedding and contrastive score computation. The latter is implemented with either the rule-based (RB) approach or the small language model (SLM) as discussed in L1008-1025 of supplementary.
>
> These experiments, support the following conclusions. First, while the selective prediction (SP) performance of $\textbf{\texttt{MA-PaPSP}}$ is not affected by the use of different retrieval methods, the computation can be substantially improved. Using ScaNN reduces latency by more than 12ms, i.e. to about $75\%$ of the kNN time. Second, for context, we compare the per-sample inference latency of $\textbf{\texttt{MA-PaPSP}}$ and several LVLMs (LVLM-as-judge). $\textbf{\texttt{MA-PaPSP}}$ has latency comparable to that of 7B parameter LVLM, but much better performance. In fact, it has substantially better performance than the 72B parameter LVLM, which is $32\times$ slower. We believe that this shows that $\textbf{\texttt{MA-PaPSP}}$ is a very computationally efficient approach to selective prediction. We will also incorporate these new results in the main paper.
>
> Regarding storage, as mentioned in L911 of the supplement, $\textbf{\texttt{MA-PaPSP}}$ requires 2.27KB per sample. For a dataset like CC12M, this has a total cost of approximately 27GB. Compressing it with Zip and Parquet can reduce the size up to 10GB and 4GB respectively. Similarly to the retrieval methods discussed above, there are various techniques to make retrieval more memory effective. For example, hashing techniques [3, 4, 5, 6] use a neural network to pre-compute a set of retrieval proxies for the dataset. The set of nearest neighbors of each proxy is then identified. During retrieval, the neural network maps the query to the closest proxy and the retrieval operation is limited to its nearest neighbors. This reduces the memory complexity of the search operation to the size of this nearest neighbor set. We have not implemented these techniques because the size of the memory required by even CC12M was not a challenge for us, even though we are an academic group with fairly limited resources. Note, for example, that it is a much bigger challenge to use LVLMs, which require higher end GPUs, than any storage costs posed by $\textbf{\texttt{MA-PaPSP}}$. In summary, when compared to something like LVLM-as-judge, $\textbf{\texttt{MA-PaPSP}}$ really is a lightweight solution both in terms of latency and infrastructure costs. It also achieves much better SP performance.
>
> > References
> 1. Stephen M Omohundro. Five balltree construction algorithms. 1989.
> 2. Mohammad Saleh Refahi et. al. Fast and scalable gene embedding search: A comparative study of faiss and scann.
> arXiv preprint arXiv:2507.16978, 2025.
> 3. Xiaofan Zhang. Towards large-scale histopathological image analysis: Hashing-based image retrieval. IEEE Transactions on Medical Imaging, 34(2):496–506, 2014.
> 4. Hui Cui et. al. Scalable deep hashing for large-scale social image retrieval. IEEE Transactions on image processing, 29:1271–1284, 2019.
> 5. Josiane Rodrigues et. al. Deep hashing for multi-label image retrieval: a survey. Artificial Intelligence Review, 53(7):5261–5307, 2020.
> 6. Weihao Kong et. al. Manhattan hashing for large-scale image retrieval. In Proceedings of the 35th international ACM SIGIR conference on Research and development in information retrieval, pp.45–54, 2012.

---

> ### Author Response · Authors · 2025-11-20
> **Response to reviewer tRii (Part 2/3): distributional similarity between retrieval and evaluation datasets.**
>
> > **Q2. Was any analysis conducted to quantify the distributional similarity between the memory banks and the target evaluation datasets?**
>
> Thank you very much for your question! We are not entirely clear on the reviewer’s interpretation of "distributional similarity," and would greatly appreciate it if the reviewer could provide further clarification. In the meantime, we have referenced similar experiments in our paper and included additional experiments conducted for the rebuttal.
>
> We report, in the paper, on experiments that study how the SP performance of $\textbf{\texttt{MA-PaPSP}}$ depends on the similarity of the target evaluation dataset and the retrieval set. As defined in L351-356 of the paper, we considered the use of a random, an in-domain, an out-of-domain, and a mixed retrieval dataset. Results are shown in Table 3 and discussed in L404-423 of the main paper. These experiments show that $\textbf{\texttt{MA-PaPSP}}$ performance depends on both $\textit{size}$ and $\textit{domain coverage}$ of the retrieval set. A large generic (out-of-domain) dataset performs quite well, but can be outperformed by smaller retrieval sets covering the target domain, especially for classification. If the retrieval set is small and does not cover the target domain, performance degrades. We note that the choice of retrieval set is more of an issue for open-set problems like captioning and ITM.
>
> We now provide some additional results to add insight to the question of what makes a hard example for $\textbf{\texttt{MA-PaPSP}}$. We considered the captioning task and analyzed the SigLIP similarity scores associated with SP successes and failures. An example is declared a success when $\textbf{\texttt{MA-PaPSP}}$ correctly predicts if the caption produced by the P-VLM is correct/incorrect. Otherwise, the example is declared a failure. Figure B.1 (of updated supplementary) shows that the SigLIP scores are higher for successes and lower for failures. This implies that $\textbf{\texttt{MA-PaPSP}}$ has higher difficulty when it cannot find a close neighbor to the query. This can happen even for an in-domain dataset. In fact, the figure B.1 (of updated supplementary) shows that there is no major difference between the score distributions of successes/failures when the retrieval set is in-domain versus out-of-domain. Obviously, if there is no overlap between the two domains, performance will drop more significantly. In this sense, $\textbf{\texttt{MA-PaPSP}}$ is similar to most other machine learning techniques. We show in the next point that the LVLM-as-judge approach also suffers from this problem.

---

> ### Author Response · Authors · 2025-11-20
> **Response to reviewer tRii (Part 3/3): procuring a memory bank similar to evaluation set is not a significant challenge.**
>
> > **Q3 The necessity of procuring a memory distribution that aligns well with the target domain at test-time presents a significant practical challenge.**
>
> We agree that finding the best retrieval set can be somewhat a challenge, as acknowledged in the paper (L481-485 in section 5). We disagree that this is a ${\it significant}$ challenge, or at least that it is a more significant challenge than that posed to competing approaches. To illustrate this, we compare the SP performance of $\textbf{\texttt{MA-PaPSP}}$ and LVLM-as-judge when there is a domain misalignment in Table 3 (below), where the target domains are medical imaging datasets. These are likely to be out-of-domain for both  $\textbf{\texttt{MA-PaPSP}}$ (using SigLIP as SP-VLM and a generic retrieval set like CC12M) and most baselines.  A few conclusions can be  drawn. First, for  $\textbf{\texttt{MA-PaPSP}}$, it is much more important to use a retrieval set from the target domain than a SP-VLM of the target domain. With the domain retrieval sets, SigLIP performs nearly as well as BioMedCLIP, which is trained on medical images. Second, for retrieval sets, "target domain" is a somewhat broad concept. As can be seen in the table, using as retrieval set CheXpert, MIMI-CXR, or both has similar performance, independently of the target dataset. This suggests that, as long as the retrieval and target datasets have some overlap,  $\textbf{\texttt{MA-PaPSP}}$ performs well, which ${\it reduces}$ the challenge of finding the right dataset. Third, when combined with an out-of-domain retrieval set, the performance of  $\textbf{\texttt{MA-PaPSP}}$ drops to the level of the performance of the LVLM-as-judge. Note that, in these experiments, the out-of-domain retrieval set is CC12M, which most practitioners would consider clearly out of domain, even unrelated to this task. So, this is a ${\it worst~case}$ result for  $\textbf{\texttt{MA-PaPSP}}$. Even in this case,  $\textbf{\texttt{MA-PaPSP}}$ is only slightly inferior to the LVLM-as-judge.
>
> In general, however, it is clear that no method, including the much more expensive LVLM-as-judge, will perform well on clearly out-of-domain data. Hence, we would disagree with the statement that having a memory distribution is just a challenge. On the contrary, the results of Table 2 show that it also provides an ${\it opportunity}$. Consider a practitioner faced with the setting of the table. In the LVLM-as-judge paradigm, the practitioner will have to find an $\textit{in-domain LVLM}$. This is likely to be much harder than finding an $\textit{in-domain dataset}$. For example, a medical clinic is likely to have data from similar exams but no capacity to train its own LVLM. Even if there is a medical LVLM that can be used as judge, that same LVLM  can be used to caption the data collected in the clinic to produce an in-domain retrieval set. As seen in Table 1 (response 1/5 to reviewer 73qG), using  $\textbf{\texttt{MA-PaPSP}}$ with this dataset is much more efficient than using the LVLM-as-judge solution. This shows that the use of a memory bank is also an ${\it advantage}$ of  $\textbf{\texttt{MA-PaPSP}}$, making it $\textit{a much more flexible and adaptable solution}$ than, for example, the LVLM-as-judge, without loss of performance  in comparable settings.
>
> | Retrieval Set    | SP-VLM               | CheXpert          |               | MIMIC-CXR        |               |
> |------------------|----------------------|-------------------|---------------|------------------|---------------|
> |                  |                      | Cider-N           | Meteor        | Cider-N          | Meteor        |
> | **$\textbf{\texttt{MA-PaPSP}}$**      |                      |                   |               |                  |               |
> | CheXpert         | SigLIP$_{\text{SO-400M}}$ | 0.137             | 0.276         | 0.157            | 0.326         |
> | MIMIC-CXR        | SigLIP$_{\text{SO-400M}}$ | **0.134**         | **0.215**     | 0.143            | 0.292         |
> | Mixed            | SigLIP$_{\text{SO-400M}}$ | 0.136             | **0.217**     | **0.132**        | **0.272**     |
> | | | | | | |
> | CheXpert         | BioMedCLIP            | **0.126**         | **0.198**     | 0.142            | 0.216         |
> | MIMIC-CXR        | BioMedCLIP            | 0.138             | 0.208         | **0.124**        | **0.202**     |
> | Mixed            | BioMedCLIP            | 0.136             | 0.204         | 0.138            | 0.268         |
> | Out-of-domain| SigLIP$_{\text{SO-400M}}$ | 0.252             | 0.326         | 0.275            | 0.276         |
> | **LVLM-as-judge**    |
> |                  | Qwen-2.5-VL (72B)     | 0.244             | 0.312         | 0.238            | 0.264         |
> |                  | Qwen-2.5-VL (7B)      | 0.248             | 0.324         | 0.268            | 0.274         |
>
> **Table 3:** AURC ($\downarrow$) for biomedical models and datasets.

---

### Official Review · Reviewer_73qG · 2025-10-31

**Soundness:** 3
**Presentation:** 3
**Contribution:** 3
**Rating:** 4
**Confidence:** 3

**Summary:**

This paper addresses the problem of selective prediction (SP) for VLMs, extending it to open-set tasks like image captioning where the output space is unbounded. The authors propose a training-free, "plug-and-play" method called Memory Augmented Plug-and-Play Selective Prediction (MA-PaPSP). This approach uses an external VLM as a "judge" to score the output of a primary predictor VLM. To overcome the inherent instability and poor calibration of the judge VLM's embeddings, MA-PaPSP is augmented with a retrieval dataset. This memory is leveraged to compute more stable "proxy embeddings" via nearest-neighbor averaging and to derive well-calibrated "contrastive scores" by comparing the prediction against a set of hard negatives. The primary contribution is a practical, lightweight framework for enabling VLMs to abstain from low-confidence predictions without requiring any model retraining.

**Strengths:**

- The focus on creating a training-free, plug-and-play solution that is applicable to open-set tasks like captioning is a valuable direction for improving the safety and reliability of modern VLMs.
- While the constituent components (selective prediction, memory augmentation, contrastive scoring) are not new in isolation, their combination to address open-set SP for VLMs in a training-free manner is a simple and well-motivated approach.
- The proposed method is technically sound and is validated through a comprehensive set of experiments. The authors test their approach across a reasonable taxonomy of tasks and demonstrate clear improvements over baselines.

**Weaknesses:**

1. **Omission of Inference Cost Analysis:** The claim of being "light-weight" is not substantiated with empirical evidence. The method introduces a potentially computationally expensive k-NN search over a large-scale dataset (e.g., CC12M) and the storage requirements for the embeddings. The paper lacks analysis of latency, throughput, or memory footprint, which is a critical omission for a method proposed for practical application.
2. **Methodological Limitations and Dependencies:**
    1. **Performance is Fundamentally Capped by the Judge VLM:**  The method's ability to assess prediction quality is inherently limited by the semantic understanding of the external "judge" VLM (e.g., SigLIP). While the proposed techniques address the *structure* of the embedding space (instability, calibration), they cannot correct for the judge's fundamental semantic blind spots, such as failures in counting, fine-grained spatial reasoning, or attribute binding. This upper bound on performance is an important limitation that is not adequately discussed.

    2. **Ignores Predictor's Internal Confidence:** The method's design completely disregards any internal confidence signals from the predictor VLM. While this enables the plug-and-play functionality, it also discards a potentially valuable source of information. It is possible that a hybrid approach, combining the external MA-PaPSP score with the predictor's internal state (when accessible), could yield superior performance.
3. **Potential Fragility from Distribution Shift and Heuristics:**
    1. There is a potential distribution shift between the captions generated by modern, powerful VLMs (often long and detailed) and the typically short, noisy captions found in web-scrapped datasets like CC12M. This mismatch could compromise the quality of the proxy embedding.
    2. The effectiveness of the contrastive score relies heavily on the quality of the "hard negatives." The paper states these are generated via a rule-based or small LM approach, which appears heuristic and may be a source of fragility. The sensitivity of the method to this generation process is not explored.
4. **Disorganized Structure and Presentation:**
    1. The structure of the experiment section (Section 4) hinders readability. The mixing of dataset descriptions, baseline details, main results, and ablation studies makes the paper's empirical contributions difficult to follow.
    2. Minor but persistent presentation issues, such as very small font sizes in figures (especially Figure 2,3,4,5) hinders readability.

**Questions:**

1. Could you provide a quantitative analysis of the inference costs (e.g., latency in ms/sample, memory usage) introduced by MA-PaPSP, especially when using a large retrieval set like CC12M? A comparison to the baseline inference time of the predictor VLM would be very helpful to contextualize the "light-weight" claim.
2. The text distribution of captions from modern VLMs like Qwen-2.5-VL differs significantly from the web-scraped captions in CC12M. How robust is the proxy embedding calculation to this domain gap? Have you investigated whether this shift negatively impacts the performance for state-of-the-art predictor models?
3. The proposed framework seems difficult to apply directly to multiple-choice Visual Question Answering (VQA) tasks, where the model's output is often a single character (e.g., 'A', 'B') whose text embedding is semantically meaningless. Could you clarify how the method could be adapted for such a common VLM task, or is this considered outside the scope of the current work?
4. The decision to completely ignore the predictor VLM's internal confidence is a core design choice. Have you considered or experimented with scenarios where this signal is available (e.g., for non-black-box models)? Could combining the MA-PaPSP score with a measure of the predictor's internal confidence lead to further improvements?

---

> ### Author Response · Authors · 2025-11-21
> **Response to reviewer 73qG (Part 1/5): Quantitative analysis of inference costs**
>
> We are pleased to learn that you recognize the significance of the problem addressed in our work, appreciate the novelty of our approach, and find our analysis to be comprehensive with clear improvements on all of our baselines.
>
> Below, we have addressed the queries you raised and hope that our responses fully clarify your concerns. If you have any more questions, please don't hesitate to let us know. We will try to promptly answer them as well.
>
> > **Q1. Could you provide a quantitative analysis of the inference costs (e.g., latency in ms/sample, memory usage) introduced by MA-PaPSP, especially when using a large retrieval set like CC12M?**
>
> This is a non-trivial question because the computation overhead can be significantly reduced by using various techniques that are popular in the retrieval literature but not the focus of the paper. We provided some discussion in the paper supplement, $\textit{e.g.}$ Table B.5 and discussion starting in L890. However, all results reported to the implementation of retrieval with k-NN and the classic Ball Tree algorithm [1], which is not close to the state of the art in terms of computation. We now present a comparison using ScaNN [2], which is a recent deep-learning based approach to retrieval, and pipeline parallelism across 4 RTX-A6000 GPUs, which is a commonly used procedure in practical applications of retrieval. Detailed results for this setting are summarized in Table 1. All the $\textbf{\texttt{MA-PaPSP}}$ methods implement both the proxy embedding and contrastive score computation. The latter is implemented with either the rule-based (RB) approach or the small language model (SLM) as discussed in L1008-1025 of supplementary.
>
> | Methods                        | Time (ms) ↓ | MS-COCO | Flickr  |
> |--------------------------------|-------------|---------|---------|
> | **$\textbf{\texttt{MA-PaPSP}}$ with k-NN retriever** |             |         |         |
> | $\textbf{\texttt{MA-PaPSP}}$-L (RB)                 | 49.54       | 0.109   | 0.219   |
> | $\textbf{\texttt{MA-PaPSP}}$-L (SLM)                | 58.85       | 0.103   | 0.216   |
> | **$\textbf{\texttt{MA-PaPSP}}$ with ScaNN retriever**|             |         |         |
> | $\textbf{\texttt{MA-PaPSP}}$-L (RB)                 | 37.54       | 0.109   | 0.219   |
> | $\textbf{\texttt{MA-PaPSP}}$-L (SLM)                | 45.85       | 0.103   | 0.214   |
> | **$\textbf{\texttt{PaPSP}}$ with LVLM-as-judge**   |             |         |         |
> | Qwen-2.5-VL (72B)              | 1563.82     | 0.115   | 0.227   |
> | Qwen-2.5-VL (7B)               | 49.38       | 0.120   | 0.233   |
> | Qwen-2.5-VL (3B)               | 27.64       | 0.147   | 0.259   |
>
> **Table 1:** Average inference times and performance of different $\textbf{\texttt{MA-PaPSP}}$ models on captioning with out-of-domain retrieval set. $\textbf{\texttt{MA-PaPSP}}$-L means $\textbf{\texttt{MA-PaPSP}}$ implemented with SigLIP-SO400M.
>
> These experiments, support the following conclusions. First, while the selective prediction (SP) performance of $\textbf{\texttt{MA-PaPSP}}$ is not affected by the use of different retrieval methods, the computation can be substantially improved. Using ScaNN reduces latency by more than 12ms, i.e. to about $75\%$ of the kNN time. Second, for context, we compare the per-sample inference latency of $\textbf{\texttt{MA-PaPSP}}$ and several LVLMs (LVLM-as-judge). $\textbf{\texttt{MA-PaPSP}}$ has latency comparable to that of 7B parameter LVLM, but much better performance. In fact, it has substantially better performance than the 72B parameter LVLM, which is $32\times$ slower. We believe that this shows that $\textbf{\texttt{MA-PaPSP}}$ is a very computationally efficient approach to selective prediction. We will also incorporate these new results in the main paper.
>
> Regarding storage, as mentioned in L911 of the supplement, $\textbf{\texttt{MA-PaPSP}}$ requires 2.27KB per sample. For a dataset like CC12M, this has a total cost of approximately 27GB. Compressing it with Zip and Parquet can reduce the size up to 10GB and 4GB respectively. Similarly to the retrieval methods discussed above, there are various techniques to make retrieval more memory effective. For example, hashing techniques [3, 4, 5, 6] use a neural network to pre-compute a set of retrieval proxies for the dataset. The set of nearest neighbors of each proxy is then identified. During retrieval, the neural network maps the query to the closest proxy and the retrieval operation is limited to its nearest neighbors. This reduces the memory complexity of the search operation to the size of this nearest neighbor set. We have not implemented these techniques because the size of the memory required by even CC12M was not a challenge for us, even though we are an academic group with fairly limited resources. Note, for example, that it is a much bigger challenge to use LVLMs, which require higher end GPUs, than any storage costs posed by $\textbf{\texttt{MA-PaPSP}}$.

---

> ### Author Response · Authors · 2025-11-21
> **Response to reviewer 73qG (Part 2/5): Robustness of the proxy embedding calculation**
>
> > **Q2. The text distribution of captions from modern VLMs like Qwen-2.5-VL differs significantly from the web-scraped captions in CC12M. How robust is the proxy embedding calculation to this domain gap? Have you investigated whether this shift negatively impacts the performance for state-of-the-art predictor models?**
>
> We believe the reviewer missed an important detail, mentioned in L351-356: the out-of-domain retrieval dataset, $\textit{i.e.}$~CC12M+CC3M+SBU, was re-captioned for our experiments, using the BLIP-2-OV (2.7B) model. Thus, while there is some shift between its text distribution and that on which modern VLMs are trained, this shift is not major, certainly much smaller than that of the original  web-scrapped captions. Having said this, the question of the importance of this shift on $\textbf{\texttt{MA-PaPSP}}$ performance is an interesting one. We did additional experiments to test the robustness of $\textbf{\texttt{MA-PaPSP}}$ to the distribution of retrieval dataset captions. Table 2 compares various re-captioning methods, showing that the quality of the captions has little impact on $\textbf{\texttt{MA-PaPSP}}$ performance. While re-captioning with a VLM achieves some gain over the original noisy captions, the performance differences between VLMs are minor, and no VLM has consistently better performance on all tasks. In fact, while all VLMs outperform the original captions, even the differences to these are not major for most tasks. The most notable exception is classification on Flowers, where the re-captioning operation enables significant gains. For the remaining tasks, the differences are  smaller. We believe that this shows that $\textbf{\texttt{MA-PaPSP}}$ is quite robust to potential caption distribution shifts between retrieval and target datasets.
>
> | Re-captioning Model     | MS-COCO Cider-N | MS-COCO Meteor | Flickr-30K Cider-N | Flickr-30K Meteor | Flowers | Pets  | UCF101 | SugarCrepe | WinoGround | What'sUp | VL-Checklist | Foil |
> |-------------------------|-----------------|----------------|---------------------|--------------------|---------|-------|--------|------------|------------|----------|--------------|------|
> | | **Captioning**          |                 |                |                     |   **Classification** |    |   | **ITM**     |            |          |              |      |
> | Qwen-2.5-VL (7B)        | **0.108**       | **0.284**      | **0.219**           | 0.295              | **0.063** | 0.116 | 0.090 | 0.062      | **0.194**  | **0.192** | **0.192**   | 0.192 |
> | LLaVA-OV (7B)           | 0.109           | **0.284**      | 0.221               | **0.292**          | 0.065   | **0.112** | 0.090 | **0.058** | **0.194** | 0.196   | 0.194        | **0.186** |
> | BLIP-2-OV (2.7B)        | 0.109           | 0.286          | **0.219**           | 0.297              | **0.063** | 0.114 | **0.088** | 0.062    | 0.196     | **0.192** | **0.192** | 0.189 |
> | **Original (web)**      | 0.113           | 0.293          | 0.223               | 0.302              | 0.092   | 0.134 | 0.092 | 0.068      | 0.197     | 0.204   | 0.206        | 0.211 |
>
> **Table 2**: Selective prediction AURC ($\downarrow$). Performance of re-captioned out-of-domain retrieval set with different LVLMs. "Web" means retrieval datasets with their original bad quality captions.

---

> ### Author Response · Authors · 2025-11-21
> **Response to reviewer 73qG (Part 3/5): Combining internal confidence scores will not lead to better performance.**
>
> > **Q3. The method's design completely disregards any internal confidence signals from the predictor VLM. While this enables the plug-and-play functionality, it also discards a potentially valuable source of information. Could combining the MA-PaPSP score with a measure of the predictor's internal confidence lead to further improvements?**
>
> The reviewer is correct that assuming no P-VLM scores is critical for the plug-and-play functionality of $\textbf{\texttt{MA-PaPSP}}$. Three difficulties arise when trying to leverage P-VLM scores. First, plug-and-play methods must, by definition, be applicable to black-box P-VLMs, for which confidence signals cannot be expected. Second, confidence signals are usually also not produced by white-box VLMs. Even for probability based VLMs, e.g. LVLMs, the probabilities are usually not sufficiently well calibrated to be meaningful. Hence, for most white-box P-VLMS, a serious effort is usually required to produce or calibrate probabilities. This can be seen from the popularity of schemes like $\textbf{\texttt{PaPSP}}$ (e.g. using CLIP as a judge [7] or LVLM-as-judge [8, 9], which reflects the fact that the confidence scores of most P-VLMs are not usable. In fact, the challenge for "judging-based approaches" is usually to produce a confidence score that can replace the one missing from the P-VLM. Third, even if it were possible to elicit such confidence signals, combining them with $\textbf{\texttt{MA-PaPSP}}$ would likely require some sort of P-VLM specific training or adaptation layers. This defeats the plug-and-play purpose of $\textbf{\texttt{PaPSP}}$, $\textbf{\texttt{MA-PaPSP}}$, or even LLM-as-judge. The point of these approaches is to be applicable without such P-VLM specific engineering.
>
> All of this is to say that assuming P-VLM confidence scores is against the nature of plug-and-play solutions. It does not mean that there could be no benefit in leveraging such signals when they exist. However, this would have to be achieved without sacrificing the plug-and-play property of $\textbf{\texttt{PaPSP}}$. We are not aware, at this point, of an easy way to do this. For applications of narrower scope, using specific P-VLMS, it would likely be beneficial to combine $\textbf{\texttt{MA-PaPSP}}$ and P-VLM confidence signals, $\textit{e.g}$ by training adaptation layers or using some probability calibration technique. This is something that we have not yet pursued because it conflicts with the plug-and-play functionality, which is the main focus of this paper.
>
> > **Q4. The paper states ``hard negatives" are generated via a rule-based or small LM approach, which appears heuristic and may be a source of fragility. The sensitivity of the method to this generation process is not explored.**
>
> This is not true. The reviewer missed supplementary section L1008-1024 and results of Table 4 in the paper, which present results of a study of the sensitivity of $\textbf{\texttt{MA-PaPSP}}$ to the process used to generate negative samples. The table shows that the use of negative samples improves the performance of $\textbf{\texttt{MA-PaPSP}}$ over not using them. However, the exact implementation does not make a significant difference. We have tried a rule-based approach, small LMs and large LMs. There is some improvement in performance when the SLM or LLM are used but it is quite small. On the other hand, these methods increase the inference time significantly.  Hence, for most applications, the use of LMs is not justified. The rule-based contrastive component is computationally more effective and nearly as effective in terms of SP performance. This happens despite the fact that the LM-generated captions are generally more coherent and grammatically correct as shown in Figure E.2 of the supplement. These results clearly show that there is no fragility associated with the contrastive component of $\textbf{\texttt{MA-PaPSP}}$. The negative sentences do not even need to be fully coherent (see e.g. the example "a boat sitting on a school on a rainbow" in Figure E.2) to enable improved performance by $\textbf{\texttt{MA-PaPSP}}$. This is because they are just used as negatives for the contrastive component and a common occurrence in contrastive learning, where the use of very weakly defined "negatives," frequently even incoherent, enables a very robust learning mechanism.

---

> ### Author Response · Authors · 2025-11-21
> **Response to reviewer 73qG (Part 4/5): Prediction quality limited by SP-VLM.**
>
> > **Q5. The method's ability to assess prediction quality is inherently limited by the semantic understanding of the external "judge" VLM (e.g., SigLIP).**
>
> This is a true statement for any approach based on an external judge. It is also true for any method that does not use an external judge. In that case, the method is inherently limited by the semantic understanding of the P-VLM itself. The important question is whether using the external judge has weaker performance than methods that do not use it. Table 2 in the paper shows clearly that this is not the case. The table just confirms what many others observed and the reason why there is so much use of LVLM-as-judge approaches in the literature. Having said this, it could be that $\textbf{\texttt{MA-PaPSP}}$ ability is limited by the use of an SP-VLM like SigLIP instead of more complex judges, like an LVLM. Our results show clearly that this is not true, i.e. the use of an SP-VLM is $\textit{not}$ a bottleneck for $\textbf{\texttt{MA-PaPSP}}$. We specifically tested this, see $\textit{e.g.}$ the results presented in the supplementary material L837–844. Essentially, we compared the performance of $\textbf{\texttt{MA-PaPSP}}$ to that of the LVLM-as-judge method, which is becoming increasingly popular for the evaluation of captioning results when complexity is not an issue. Table B.4 of the supplement shows that $\textbf{\texttt{MA-PaPSP}}$ has superior SP performance than LVLM-as-judge, across various LVLM sizes.
>
> We now extend this comparison in Table 1 of this rebuttal, to also include latencies. The table shows that $\textbf{\texttt{MA-PaPSP}}$ with SigLIP (a 1B-parameter model) as the SP-VLM is much faster $\textit{and}$ achieves better performance than much larger LVLMs, such as Qwen-2.5-VL (72B parameters). This shows that the SP-VLM is not a bottleneck. This finding is justified by the fact that the vision encoders of most LVLMs are quite similar to the SP-VLMs that we use. So, LVLMs suffer from exactly the same issues that the \ours model faces. Memory augmentation with equation 6 and contrastive captions with equation 7 improves the representation of the SP-VLM and help overcome these issues. This type of improvement is difficult to achieve for LVLMs, where the encoder is inside the model and not accessible. Improving the encoder requires retraining the whole model. In any case, as the visual encoders of LVLMs improve, those encoders can themselves be used as SP-VLMs in $\textbf{\texttt{MA-PaPSP}}$. So, it is more likely that SP-VLMs will advance faster than LVLMs. In fact, we leverage this, e,g, by using the more recent SigLIP instead of CLIP as SP-VLM. Many LVLMs are still implemented with CLIP, which is a weaker representation model but not easy to replace. Our results show that $\textbf{\texttt{MA-PaPSP}}$ is quite effective even for the state-of-art SigLIP representation model. To the best of our knowledge, there is no method in the literature whose SP performance is superior to $\textbf{\texttt{MA-PaPSP}}$, either using an external judge or not.
>
> > **Q6. While the proposed techniques address the structure of the embedding space (instability, calibration), they cannot correct for the judge's fundamental semantic blind spots, such as failures in counting, fine-grained spatial reasoning, or attribute binding. This upper bound on performance is an important limitation that is not adequately discussed.**
>
> We strongly disagree with this statement. The judge used by $\textbf{\texttt{MA-PaPSP}}$ does not need to be able to count by itself, only needs to be able to retrieve from the retrieval set the relevant counting examples. A P-VLM caption ``This is an image with 5 apples'', will likely induce the retrieval of various image-text pairs containing 5 objects from the retrieval set. These are likely to be more similar to the image being captioned if the latter has five objects than if it does not. So, $\textbf{\texttt{MA-PaPSP}}$ can solve the problem even if the SP-VLM "cannot count". Our results confirm this. The ITM task, which we use throughout the paper, was actually developed to provide clarity about problems like these. The task is to choose the correct captions among multiple "negative captions" for an image. These negative captions differ from the positive caption in terms of counting, fine-grained spatial reasoning, attribute binding, and other "fine-grained" details. Please see the above references for more details on these datasets. Tables A.1, B.1, B.2, B.3 and B.4 show that $\textbf{\texttt{MA-PaPSP}}$ outperforms the baselines on all these datasets.

---

> ### Author Response · Authors · 2025-11-21
> **Response to reviewer 73qG (Part 5/5): MA-PaPSP can handle MC-VQA**
>
> > **Q7. The proposed framework seems difficult to apply directly to multiple-choice Visual Question Answering (VQA) tasks, where the model's output is often a single character (e.g., 'A', 'B') whose text embedding is semantically meaningless.**
>
> We strongly disagree with this statement. The VQA task never consists of answers that are simply 'A', 'B', or 'C'. There is always some text associated with these options, $\textit{e.g.}$ 'A - 5 apples', 'B - 3 oranges'. $\textbf{\texttt{MA-PaPSP}}$ can simply use the text corresponding to each option for the score computation. We have used this procedure in all the experiments that require choosing one out of many options. For classification, where one or a few word are usually the answer, e.g. "golden retriever", we prepend a dummy sentence to the word(s) $\textit{e.g.}$ "An image of golden retriever, a dog", as is commonly done in the literature.
>
> > **Q8. Paper structure**
>
> We appreciate the reviewer for highlighting this issue. In response, we are revising the paper to improve the organization of the experimental sections and address issues such as increasing the font size to enhance readability.
>
> References -
>
> 1. Stephen M Omohundro. Five balltree construction algorithms. 1989.
> 2. Mohammad Saleh Refahi et. al. Fast and scalable gene embedding search: A comparative study of faiss and scann.
> arXiv preprint arXiv:2507.16978, 2025.
> 3. Xiaofan Zhang. Towards large-scale histopathological image analysis: Hashing-based image retrieval. IEEE Transactions on Medical Imaging, 34(2):496–506, 2014.
> 4. Hui Cui et. al. Scalable deep hashing for large-scale social image retrieval. IEEE Transactions on image processing, 29:1271–1284, 2019.
> 5. Josiane Rodrigues et. al. Deep hashing for multi-label image retrieval: a survey. Artificial Intelligence Review, 53(7):5261–5307, 2020.
> 6. Weihao Kong et. al. Manhattan hashing for large-scale image retrieval. In Proceedings of the 35th international ACM SIGIR conference on Research and development in information retrieval, pp.45–54, 2012.
> References -
> 7. Amit Zalcher et. al. Don’t judge before you clip: A unified approach for perceptual tasks. arXiv preprint arXiv:2503.13260, 2025.
> 8. Dawei Li et al. From generation to judgment: Opportunities and challenges of llm-as-a-judge. In Proceedings of the 2025 Conference on Empirical Methods in Natural Language Processing, pp. 2757–2791, 2025.
> 9. Sijun Tan et. al. Judgebench: A benchmark for evaluating llm-based judges. arXiv preprint arXiv:2410.12784, 2024.

---

### Meta-Review · Area_Chair_LjDU · 2026-01-07

**Summary:**

The paper introduces MA-PaPSP, a training-free framework for Selective Prediction (SP)—the ability for a model to abstain from low-confidence predictions. While SP is well-studied for closed-set tasks, this work extends it to open-set tasks like image captioning by using an external Memory-Augmented judge . The proposed method addresses representation instability and poor calibration in Vision-Language Models (VLMs) by leveraging proxy embeddings and contrastive scoring. The contrastive scoring is designed to provide stable calibration of confidence scores which is crucial for SP.

The initial reviews were split (ratings of 6, 4, 4, 4), primarily due to concerns regarding inference latency, the dependency on retrieval set quality, and whether a smaller SP-VLM (like SigLIP) acts as a bottleneck for larger predictors (like Qwen-2.5-VL). However, the rebuttal provided significant empirical evidence showing that MA-PaPSP actually outperforms much larger "LLM-as-judge" models (e.g., 72B parameters) while being significantly faster (approx. 40ms vs. 1500ms). The authors also demonstrated generalization to specialized domains like medical imaging by swapping the retrieval memory. Regarding the concern on converage, in experiments the author demonstrated that when the retrieval set is off-topic, the performance seems to be not show catastrophic break but instead degrades to non memory augmented levels. This type of a graceful fails is good for retrieval based systems.

Overall the meta reviewer feels the method proposed is sound and the rebuttal adequate. Therefore it is recommended to accept this work to ICLR.

**Reviewer Concerns:**

Addressed Concerns:

Inference Costs & Latency (73qG, tRii): The authors provided new quantitative data using ScaNN retrieval, demonstrating latencies of 37-45ms. In general the retrieval time is minimal compared with the seconds-level time required by VLMs to do captioning.

SP-VLM as a Bottleneck (8eZN, 73qG): Authors showed that a 1B SigLIP model with memory augmentation outperforms a 72B Qwen-2.5-VL as a judge. Also it is good to point out that most visual encoders in VLMs are of similar scale as SigCLIP.

Hard Negative Heuristics (73qG): The authors pointed to sensitivity studies in the supplement showing that even simple rule-based negatives significantly improve calibration without the fragility suggested by the reviewer.

Outstanding Concerns
Presentation & Readability (WFx9, 73qG): The authors are encouraged to revise the manuscript carefully.

Semantic Blind Spots (73qG): A reviewer remained concerned that the judge cannot correct for fundamental reasoning failures (e.g., counting) if the SP-VLM itself lacks that capability. The meta-review feels this will likely be true in many cases. And investigation into this issue is encouraged.

**Reviewer Scores:**

Reviewer tRii would likely maintain the scoring of 6.

Reviewer 8eZN would likely slightly raise the rating to 5.

Reviewer 73qG would likely keep the rating.

Reviewer WFx9  would likely keep the rating due to concerns on the novelty of the work.

---

### Decision · Program_Chairs · 2026-01-26

Accept (Poster)